# Spatial and temporal analysis of fatal off-piste and backcountry avalanche accidents in Austria with a comparison of results in Switzerland, France, Italy and the United States.

Christian Pfeifer, Dr.

Institute of Basic Sciences in Engineering Science

Unit for Engineering Mathematics, University of Innsbruck

Peter Höller, Dr.

Austrian Research Centre for Forests

Institute for Natural Hazards, Innsbruck

Achim Zeileis, Dr.

Department of Statistics, University of Innsbruck

**Address for correspondence:**

Christian Pfeifer

Institut für Grundlagen der Technischen Wissenschaften

Arbeitsbereich für Technische Mathematik, Universität Innsbruck

Technikerstraße 13, A–6020 Innsbruck

E-Mail: christian.pfeifer@uibk.ac.at

## Abstract

In this article we analyzed spatial and temporal patterns of fatal Austrian avalanche accidents caused by backcountry and off-piste skiers and snowboarders within the winter periods 1967/68–2015/16. The data were based on reports of the Austrian Board for Alpine Safety and reports of the information services of the federal states.

Using the date and the location of the recorded avalanche accidents we were able to carry out spatial and temporal analyses applying generalized additive models and Markov random field models.

As the result of the trend analysis we noticed an increasing trend of backcountry and off-piste avalanche fatalities within the winter periods from 1967/68 to 2015/16 (although slightly decreasing in recent years), which is in contradiction to the widespread opinion in Austria that the number of fatalities is constant over time. Additionally, we compared Austrian results with results of Switzerland, France, Italy and the United States based on data from the International Commission of Alpine Rescue (ICAR). As the result of the spatial analysis we noticed two hotspots of avalanche fatalities ('Arlberg-Silvretta' and 'Sölden').

Because of the increasing trend and the rather 'narrow' regional distribution of the fatalities consequences on prevention of avalanche accidents were highly recommended.

**Keywords:** Snow, Avalanches, Accidents

# 1 Introduction

In the Alps, backcountry skiing has become very popular in the last 50 years. Unfortunately, there are a lot of fatal accidents due to snow avalanches caused by skiers and/or snowboarders. They are of special public interest (Januskovecz, 1989).

In Austria, about 25–30 fatalities caused by snow avalanches are expected every year (Neuhold, 2012; Höller, 2009). Furthermore, it is reported that in Alpine countries (such as Austria) the number of fatalities is more or less constant over the time (Brugger et al., 2001; Valt & Pivot 2013; Roth, 2013) and that there is some sort of seasonality in the data in terms of higher frequencies of accidents within a distance of 5 or 6 years (Höller, 2009; Tschirky et al., 2000). Harvey and Zweifel (2008) even denote that fatalities are decreasing over time in Switzerland. In a recent paper Techel et al. (2016) investigated avalanche fatalities in the European Alps (in addition to Switzerland–Austria–Slovenia) over time stratified for controlled and uncontrolled terrain, concluding that in the case of uncontrolled terrain the trend seems to be constant over time from the 1980s up to now.

Usually trend information for Austrian avalanche fatalities is given in the annual reports of the Austrian Board for Alpine Safety (Kuratorium für alpine Sicherheit, 2016). Considering these profiles, we notice higher frequencies of fatalities in the 1980s. However, the highest frequency in winter 1998/99 is due to avalanche fatalities in villages (Galtür, Ischgl), also affecting buildings. This is because the statistics in the reports do not distinguish between fatal avalanches in buildings, on roads, outdoors without skiing, fatalities due to skiing on slopes and backcountry skiing.

In this paper our focus is on accidents caused by backcountry (using no ascent support) and off-piste ('leaving the ski resort in order to travel in areas that were not controlled for avalanches', see (Silverton et al., 2009)) skiers or snowboarders. We addressed this special group of accidents (backcountry and off-piste avalanche fatalities up to 2010/11) for the first time in a short paper, see Pfeifer et al. (2013). Most recently, Höller carried out an investigation of backcountry and off-piste avalanche fatalities stating that there is no significant trend (but a slight change) in the number of deaths (Höller, 2017).

Our task in this paper is to carry out a spatial and temporal analysis, identifying (potentially nonlinear) trends over time and regional patterns. In the case of trend analysis, we compare Austrian results with results of Switzerland, France, Italy and the United States.

## 2 Materials and methods

### 2.1 Data

For our study we built a data base of fatal avalanche accidents recording the

1. date,

2. municipal area where the accident took place,

3. federal state of the municipality,

4. number of persons involved,

5. number of fatalities,

6. type of activity (on/off-piste, backcountry skiing, etc.)

of fatal accident events in Austria within the winter periods
1980/81–2015/16, which are available from the annual reports of the
Austrian Board for Alpine Safety (Kuratorium für alpine Sicherheit, 2016)
and the annual reports of the information services of the federal states
(Amt der Tiroler Landesregierung, 2009). In order to check the reliability of
the accident data, we made a cross-check between those reported in the two
sources. Looking at winter season 1986/87 we figured out that the reports
were incomplete. However, we were able to fill this gap using records of the
BFW (Austrian Research Centre for Forests, Institute for Natural Hazards,
Innsbruck), e.g. see (Schaffhauser et al., 1988).

For the period 1967/68–1979/80 we used aggregated information pub-
lished in the annual reports of the Austrian Board for Alpine Safety (Ku-
ratorium für alpine Sicherheit, 2016). Starting from 1977/78 we were able
to distinguish between backcountry and off-piste fatalities. Finally, further
annual reports of the BFW were helpful in order to resolve classification
problems of avalanche events.

Keeping in mind aspects of data quality, it seems to be that avalanche
information back to the period 1967/68 is reliable for our purposes. In general
information relating to fatal avalanches seems to be much more reliable than
information relating only to avalanches with injured or uninjured persons.
Most notably, in the case of fatal avalanches we do not expect that there are
records missing.

In order to compare Austrian results with international results we use
data from the International Commission of Alpine Rescue (ICAR) which

was kindly made available for us by the ICAR.

The data are annual count data of fatal avalanche events ('Statistique d'accidents d'avalanche') based on 21 countries within the periods 1983/84–2015/16 which are categorized by the type of fatalities (backcountry skiing or snowboarding, off-piste, on-piste, alpinist without ski/snowboard, on road, buildings, snowmobile, other).

In case of the international data (Switzerland: Frank Techel, 'Auszug aus der Lawinenschadensdatenbank des SLF' (SLF 2017); Italy: Mauro Valt, Associazione Interregionale Neve e Valanghe, Trento; France: Frederic Jarry, ANENA; United States: Ethan Greene, Colorado Avalanche Information Center) a crosscheck was carried out.

For looking at the regional distribution of avalanche fatalities we built small area maps based on Austrian municipalities. For this purpose we use polygon boundaries of the small-scaled areas provided by the 'Bundesamt für Eich- und Vermessungswesen' (BEV) in a shapefile. In order to get a regional overview of the alpine terrain ($\geq 1500m$ above sea level) for discussion, we use digital elevation model (DEM) data from the BEV at an $250m$ resolution. Further on, we use data of overnight stays in the winter season 2015/16 at community level provided by the 'Statistik Austria' as an additional approach for discussion (https://www.statistik.gv.at/web_de/statistiken/wirtschaft/ tourismus/beherbergung/ankuenfte_naechtigungen/index.html).

## 2.2 Statistical methods

After aggregating the spatio-temporal data $y_{st}$ (denoting the observed fatalities at time $t$ and location $s$) in terms of location, which means summing up over the locations, $\sum_s y_{st}$, we propose the following model for capturing the trend over time:

$$\log(\mu_{\mathbf{t}}) = f(t) + x_t \qquad (2.1)$$

where $\mu_{\mathbf{t}}$ denotes the expectation of the Poisson distributed number of annual avalanche fatalities over time $t$ (in our case: winter periods). The logarithms of these values are modelled as the sum of potentially nonlinear trend function $f(t)$ and a stationary remainder $x_t$. We use the Aikake information criterion (AIC) and the Bayesian information criterion (BIC) in order to compare the constant, linear and nonlinear model (which is in our opinion the better choice than reporting pairwise comparisons of p-values for potentially non-parametric trend functions, see e.g. Venables & Ripley, 2002). To account for potential serial correlation and periodic variation in the remainder, we consider autoregressive moving-average (ARMA) effects.

After aggregating the spatio-temporal data $y_{st}$ in terms of time, which means summing up over the time, $\sum_t y_{st}$, we propose a Markov random field approach modelling the expected number of avalanche fatalities $\mu_{\mathbf{s}}$ ($s$, $s \in \{1, \ldots, S\}$, denoting the region which are municipalities in our case) as follows:

$$\log(\mu_{\mathbf{s}}) = Z\beta_s \qquad (2.2)$$

where the $S \times S$ design matrix $Z$ depends on the specific form of the spatial layout. The coefficients $\beta_s$ are conditionally Gaussian distributed (Markov random fields) according to:

$$\beta_s | \beta_{-s} \sim N\{\frac{1}{n_s} \sum_{r \sim s} \beta_r, \frac{\tau^2}{n_s}\} \qquad (2.3)$$

where $\beta_{-s}$ denotes the vector of parameters without its $s$th component, $n_s$ is equal to the number of neighboring regions with reference to region $s$, $s \sim r$ indexes all units adjacent to region $s$ and $\tau^2$ denotes a (unknown) variance parameter.

For fitting these models we use the R package mgcv (R Development Core Team, 2012; Wood, 2006) which applies the smoothing spline approach for fitting generalized additive models (GAM).

Further on, for looking at the regional distribution of avalanche fatalities (and subsequently at the regional distribution of alpine terrain and overnight stays) we build small area maps based on Austrian municipalities using the geographic information system (GIS) ArcMap. We, of course, use Markov random field estimates as described above which helps us to identify regional hot spots of avalanche fatalities.

# 3 Results

## 3.1 Temporal results

In the following, we give the plots of temporal estimated functions of avalanche fatalities at first plotting the function for Austria in total within the winter periods 1967/68–2015/16 (see Figure 1). Additionally, we plot

the trend function of exclusively off-piste fatalities starting from the winter season 1977/78 (see Figure 2). Further on, we calculate 90% confidence bands of the estimated functions in both cases as shown in the plots.

For reasons of comparison Table 1 gives the frequencies of backcountry, off-piste and total fatalities of Austria and the Austrian neighboring countries Italy and Switzerland within the winter periods 1983/84–2015/16. Additionally the off-piste percentages are reported. Furthermore, we report the results of fatalities in France, which turns out to be the country with the highest counts of fatalities in Europe, and the results of the United States, which is probably the most important country outside of Europe in terms of avalanche fatalities. For this purpose, however, we use ICAR data as described above.

For further international comparison we consider estimated functions of off-piste and backcountry avalanche fatalities (and off-piste fatalities detached) of Switzerland, France, Italy and the United States in Figures 3–6. In addition, Figure 7 shows temporal profiles for the combined data summing up the numbers of Austria (AUT), Switzerland (CHE), France (FRA) and Italy (ITA). And for discussion, we are looking at the numbers of Austrian backcountry accidents over time with more than 1 fatality (Figure 8).

Finally, the Aikake information criterion (AIC) and the Bayesian information criterion (BIC) of the constant (no trend effect), linear and nonlinear models are reported for model comparison – see Table 2. Lower AIC- and BIC-values, however, indicate significantly better fits when comparing the different models.

## 3.2  Regional results

Figures 9 and 10 show the regional distribution of fatal avalanche events (Figure 9 in total and Figure 10 off-piste only) using colored maps based on small areas, which are the Austrian municipalities in our case. The coloring, however, is based on Markov random field estimates of avalanche fatalities as described in the previous Section (deviance explained: total 91.2%, offpiste 87.1%); the number corresponding with each spatial unit in the plot is equal to the original count.

In addition to Figures 9–10, Table 3 gives a list of those municipalities with the most avalanche fatalities in Austria. Further on, we list those avalanche events in Austria with the highest counts of fatalities in Table 4 which turns out to be useful for the discussion section.

Finally, Figure 11 shows the distribution of Alpine terrain ($\geq 1500m$ above sea level) and the distribution of the overnight stays in the winter season 2016 at municipal level (restricted to the 130 municipalities with more than 100,000 overnight stays in Austria) which allows us to discuss possible reasons for the observed distribution of avalanche fatalities in Figure 9 (Pearson correlation Alpine terrain: 0.42, overnight stays: 0.62 ) and Figure 10 (Alpine terrain: 0.27, overnight stays: 0.66).

# 4  Discussion

## 4.1  Temporal analysis with an international overview

If we look at the trend function of Austria in total (see Figure 1) we notice an increasing trend having its maximum at winter period 2005/06 (1969/70: approx. 12, 2005/06 approx. 22). In recent years we, however, notice that the number of annual fatalities is slightly decreasing.

Additionally we take notice of a peak in the 1980s ranging between 1981/82 and 1987/88. But keeping in mind that increased snowfall has an essential effect on the number of accidents (Harvey, 2008; Harvey et al., 2012; Höller, 2012), increased solid precipitation in the 1980s during wintertime (Laternser & Schneebeli, 2003; Abegg, 1996) could give some evidence for this pattern.

Looking at the off-piste trend function (see Figure 2), we notice an increasing (linear) trend without any peak in the 1980s. As in the 'total' case, the off-piste fatalities are slightly decreasing from the mid 2000s on.

Lower AIC- and BIC-values (see Table 2) indicate that the nonlinear model is preferable to the constant or linear model – although in case of 'Austria off-piste' the BIC-value indicates that the linear model seems to be preferable.

Considering ARMA effects, we did not find any substantial serial correlation or any sort of periodicity in the remainder $x_t$. Further on, we notice that there is a lot of variation of the observed counts around estimated function(s).

Comparing Austrian fatal backcountry and off-piste counts within 1983/84 – 2015/16 with results of counts in Switzerland, France, Italy and

the United States (see Table 1) we notice, led by France (787 fatalities in total, 23.85 fatalities per year), the second largest number of total avalanche fatalities (680, 20.61) in Austria. Having a focus on backcountry fatalities only, Austria is leading (458, 13.88) followed by France (433,13.12) and Switzerland (395, 11.97). In Austria a share of 32.65% of total fatalities are due to off-piste accidents (largest value France: 44.98%; smallest: United States 29.23%).

Comparisons with total fatality profiles of France, Switzerland and Italy (and profiles of the summing-up of AUT, CHE, FRA and ITA) result in:

1. high frequencies in the 1980s,

2. low counts in the 1990s,

3. increasing trend beginning in 2000

4. to some extent decreasing in recent years,

which in turn is rather similar to the results of Austria.

However, if we consider the results of the United States in Figure 6 (284 total fatalities, 8.61 fatalities per year) we note a positive almost linear trend without any peaks in the 1980s. The AIC- and BIC-values indicate that, with the exception of the United States (linear model), nonlinear models are preferable (whereas the BIC-values of France almost indicate that there is no effect at all in case of France).

If we compare the off-piste trends of the countries we notice quite different shapes to those of Austria (positive trend without peak in the 1980s):

1. Italy: similar to shape as seen in case of total counts.

2. Switzerland: difference to total trend function, peak of off-piste trend around year 2000 (which is very similar to the profile of the summing-up of AUT, CHE, FRA and ITA).

3. France: decrease of off-piste counts in recent years.

4. United States: almost no increase; because of the lowest AIC-value, the constant model turns out to be the best one.

Such as in the 'total' case above, lower AIC- and BIC-values indicate that, with the exception of the United States (constant model), nonlinear models are best-performing. Usually trend information is given as a linear function in the literature for avalanche data, see e.g. (Tschirky et al., 2000; Harvey & Zweifel, 2008; Spencer & Ashley, 2011; Page et al., 1999). Our investigations - see AIC- and BIC-values in Table 2 - showed that (with the exception of the US-data) linear models are not appropriate – see also the results of (Techel et al, 2016), (SLF, 2016) and (Höller 2017) in the recent research. However, Techel (2016) and Höller (2017) could not find significant results because they were using a nonparmetric test (Mann-Kendall) which is only sensitive for linear or monotonic trend profiles.

At the beginning and the end of the longitudinal profiles we observe larger confidence bands indicating less precise estimates due to missing data in their neighbourhoods. As a result of this extreme estimates at the beginning of the temporal profiles could be less reliable (e.g. in case of Switzerland 'total', if we compare the results with those of SLF, 2016).

We think that single extreme events do not have an influence on the estimated functions because of the robustness of the estimator. In this

context, we observe a significant decrease of number of avalanche fatalities with more than 1 fatality, see Figure 8 and AIC/BIC values of the constant (142.77/144.02), linear (139.17/142.37) and nonlinear model (141.19/146.34) suggesting that the linear model is preferable.

The temporal profiles could also be seen as an indicator for low/high frequency temporal clusters, which are: Austria total (6 larger values) in the mid 1980s; Switzerland off-piste, France and Italy total (5 smaller values for each) in the early and mid 1990s.

## 4.2   Regional analysis

In Figure 9 we explore the regional or spatial distribution of avalanche fatalities in Austria within the years 1981–2016. Here the total area of Austria is divided into small areas, equal to the areas of the Austrian municipalities (211 municipalities with at least one reported fatality). Looking at Table 3, we notice that the municipalities with highest numbers are 'Sölden' and 'St. Anton a. Arlberg' Around the municipalities 'St. Anton a. Arlberg' and 'Sölden' in the western part of the Austrian federal state Tyrol we observe 2 clusters or hot spots of increased fatalities:

The first cluster (CL1), centered around the regions Arlberg and Silvretta, is including the municipalities St. Anton a. Arlberg (number of avalanche fatalities: 31), Kaisers (10), Klösterle (9), Lech (22) in Arlberg, and the municipalities St. Gallenkirch (8), Gaschurn (8), Galtür (21), Ischgl (9) in Silvretta.

The second cluster (CL2), located in the southern part of Ötztal, Kühtai and Stubai, is including the municipalities Sölden (50), St. Leonhard i. Pitztal

(18), Längenfeld (9) in the Ötztal Alps, and the municipalities St. Sigmund i. Sellrain (11), Silz (14), Sellrain (5), Neustift i. Stubaital (11) in Kühtai-Stubai.

Further on, we observe some smaller spots in the federal states:

– Tyrol (Tuxer Alpen): Navis (9), Wattenberg (9), Schmirn (5), Tux (10)

– Salzburg (Saalbach): Saalbach-Hinterglemm (10), Niedernsill (13)

– Styria (Triebener Tauern – Seckauer Tauern): Gaal (6), Wald am Schoberpaß (6), Hohentauern (6).

Finally we notice some single areas with increased frequency such as:

Mittelberg Vorarlberg (10), Heiligenblut Carinthia (11), Werfenweng Salzburg (15), Pusterwald Styria (10). Some single areas with increased frequencies, e.g. Werfenweng (15) and Niedernsill (13), are due to disastrous single avalanche events, see e.g. Table 4.

Figure 10 plots the distribution of the off-piste fatalities (without backcountry fatalities; 77 municipalities with at least one reported off-piste fatality). As a conclusion we notice 2 hot spots of off-piste fatalities which are: 'St. Anton a. Arlberg' - 'Lech' - 'Ischgl' (Arlberg, Ischgl) and 'Sölden' (southern part of Ötztal).

Furthermore, there are some single spots or small clusters such as: Tux Tyrol (5), Jochberg Tyrol (5) , Saalbach-Hinterglemm Salzburg (6), Niedernsill Salzburg (12).

If we compare Figure 9 and Figure 10 (or if we have a look at Table 3) we notice centres of off-piste avalanche fatalities in CL1 such as Lech (20 off-piste fatalities out of 22 total, 90.91% off-piste), St. Anton a. Arberg ( 26 out of 31 total, 83.87%) and Ischgl (9 out of 9 total, 100%) while the accidents

of Galtür (0% off-piste), St. Gallenkirch (2 out of 8 total) and Gaschurn (0% off-piste) are mainly due to backcountry skiers.

Looking at CL2, the fatal accidents of our interest are mainly caused by backcountry skiers except Sölden which off-piste rate is about 50% (25 out of 50 total, $\geq 32.65\%$ in case of Austria).

Figure 11 (distribution of alpine terrain and overnight stays in the winter season 2015/16) tries to give some idea in order to explain the spatial distribution of avalanche fatalities of Figure 9 and Figure 10. Obviously, the percentage of alpine terrain at municipal level coincides with the number of fatalities. However, there are alpine areas with less number of fatalities than in those in the western part of Tyrol, see e.g. East Tyrol. The majority of fatalities are restricted to 2 clusters, which is more or less only a small part of the terrain of our interest.

Looking at the overnight stays in Figure 11, we notice that the largest counts of overnight stays coincide with the largest counts of in total and off-piste fatalities (Sölden, St. Anton a. Arlberg, Lech), but there are winter tourist regions with less number of avalanche fatalities, see e.g. the Tauern region or the northeastern part of Tyrol. In the case of total number of fatalities, overnight stays are partly misleading because they e.g. do not take into account the considerable number of native backcountry skiers around Innsbruck.

This is more or less in agreement with considering the size of Austrian ski resorts, see (Fleischhacker, 2016), instead of overnight stays (Spencer and Ashley (2011) stated that areas with higher winter sports activity are those with higher number of avalanche fatalities).

Finally, if we consider the spatial patterns of buildings exposed to snow avalanches in Austria (Fuchs et al., 2015) we could find some remarkable congruences (looking at CL1 and CL2) if we compare them with avalanche fatalities at municipal level.

# 5    Conclusion

As the result of the trend analysis we notice an increasing trend (although decreasing in recent years) of off-piste and backcountry avalanche fatalities within the winter periods from 1967/68 to 2015/16. This clearly contradicts the widespread opinion that the number of fatalities is constant over time.

Comparing results of off-piste and backcountry avalanche fatalities in Austria with other relevant countries we notice the second highest number of off-piste and backcountry fatalities in Austria and the largest number of backcountry fatalities in Austria. We notice similar estimated functions if we compare Austrian results with results of the relevant European countries. However, the off-piste trend function of Austria is quite different to those of the other relevant European countries (but similar to those of the United States).

As the result of the regional analysis we notice two hot spots of avalanche fatalities in Figure 9: 'St. Anton a. Arlberg (29)' (Arlberg-Silvretta) and 'Sölden (43)' (southern part of Ötztal, Stubai-Kühtai).

Because of the increasing trend (although decreasing in recent years) and the rather 'narrow' regional distribution of the fatalities, consequences on prevention of avalanche accidents are highly recommended, e.g. starting a

‘campaign against avalanche accidents’ in the centers of the clusters St. Anton and Sölden. This should especially be done in order to prevent the large number of off-piste (freerider) fatalities in St. Anton-Lech-Ischgl and Sölden. Additionally, we observe decreasing numbers of fatal backcountry avalanches with more than 1 fatality, see Figure 8, which could be the effect of more awareness of danger in the last 30 years.

Unfortunately, we are not able to verify the influence of increased number of backcountry and off-piste skiers over time because there is no valid information about frequencies of backcountry and off-piste skiers in general. However, we find some evidence that increased winter overnight stays (which could be seen as an evidence for increased winter sports activity) has an effect on higher number avalanche fatalities, see Figure 11.

Finally, we do not hesitate to mention that further research is needed, e.g. to explore the influence of new fallen snow, temperature, etc. on the number of fatalities in a spatio-temporal model. For this purpose, further and more precise data are necessary.

**Acknowledgement:** The authors kindly acknowledge data provision from the International Commission of Alpine Rescue (ICAR) by Hans-Jürg Etter. Additionally, the authors are grateful to Frank Techel (SLF, Davos Switzerland), Mauro Valt (Associazione Interregionale Neve e Valanghe, Trento Italy), Frederic Jarry (Association Nationale pour l'Étude de la Neige et des Avalanches, Grenoble France) and Ethan Greene (Colorado Avalanche Information Center, Boulder United States) which allowed us to check the data provided from the ICAR.

Financial/Material Support: None

Disclosures: None

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

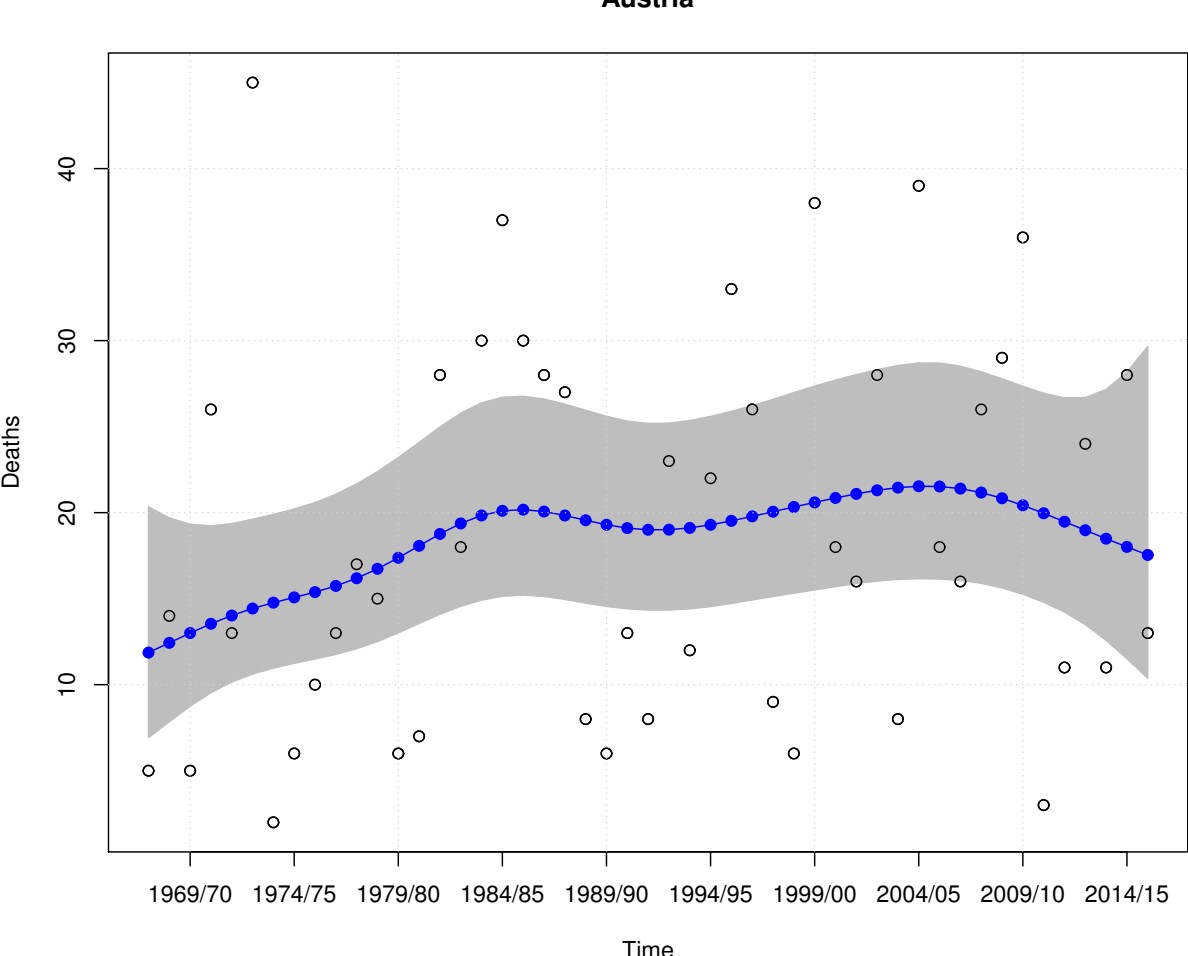

Figure 1: Observed (∘) and estimated (●) annual total avalanche fatalities (off-piste and backcountry) with 90% confidence band (grey) in Austria within 1967/68–2015/16.

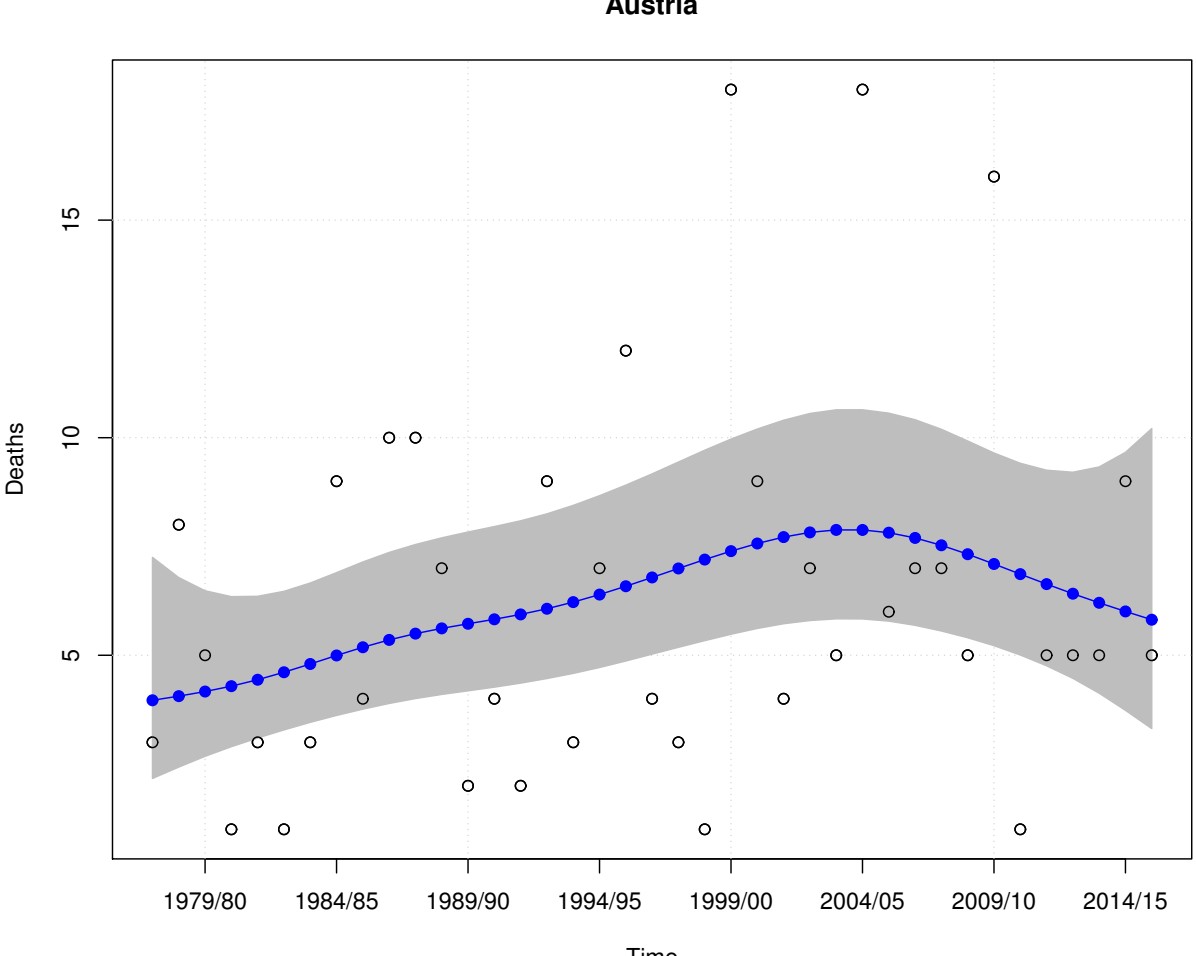

**Austria**

Figure 2: Observed (○) and estimated (•) annual off-piste avalanche fatalities with 90% confidence band (grey) in Austria within 1977/78–2015/16.

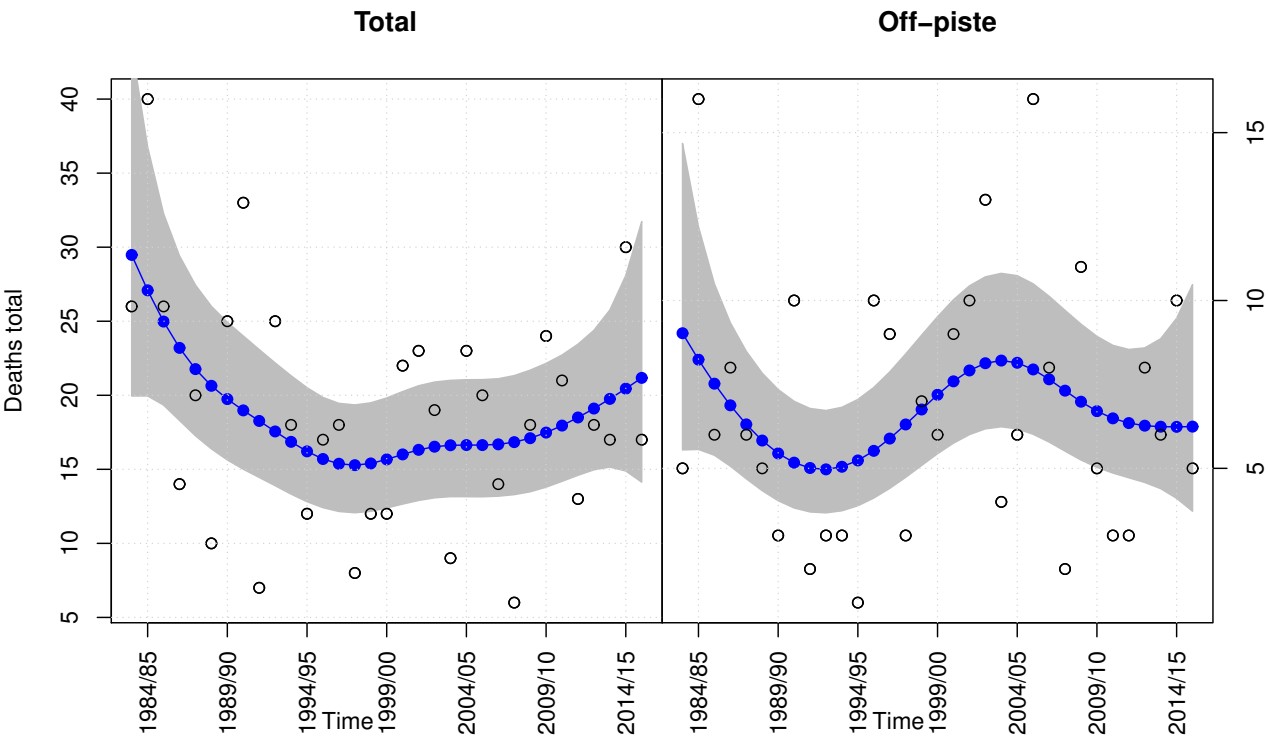

Figure 3: Observed (○) and estimated (●) annual avalanche fatalities (off-piste and backcountry, i.e. total, on the left and off-piste on the right) with 90% confidence bands (grey) in Switzerland within 1983/84–2015/16.

Figure 4: Observed (○) and estimated (●) annual avalanche fatalities (off-piste and backcountry, i.e. total, on the left and off-piste on the right) with 90% confidence bands (grey) in France within 1983/84–2015/16.

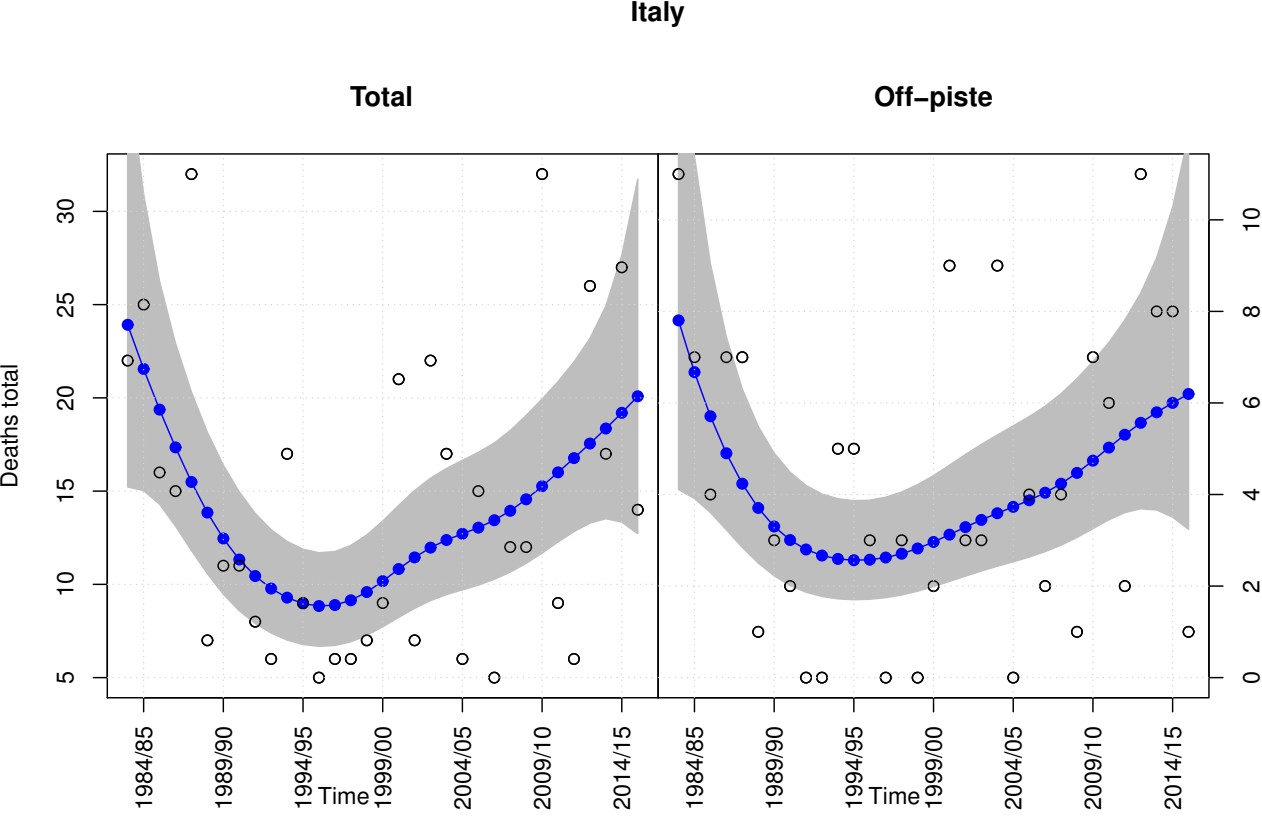

Figure 5: Observed (○) and estimated (●) annual avalanche fatalities (off-piste and backcountry, i.e. total, on the left and off-piste on the right) with 90% confidence bands (grey) in Italy within 1983/84–2015/16.

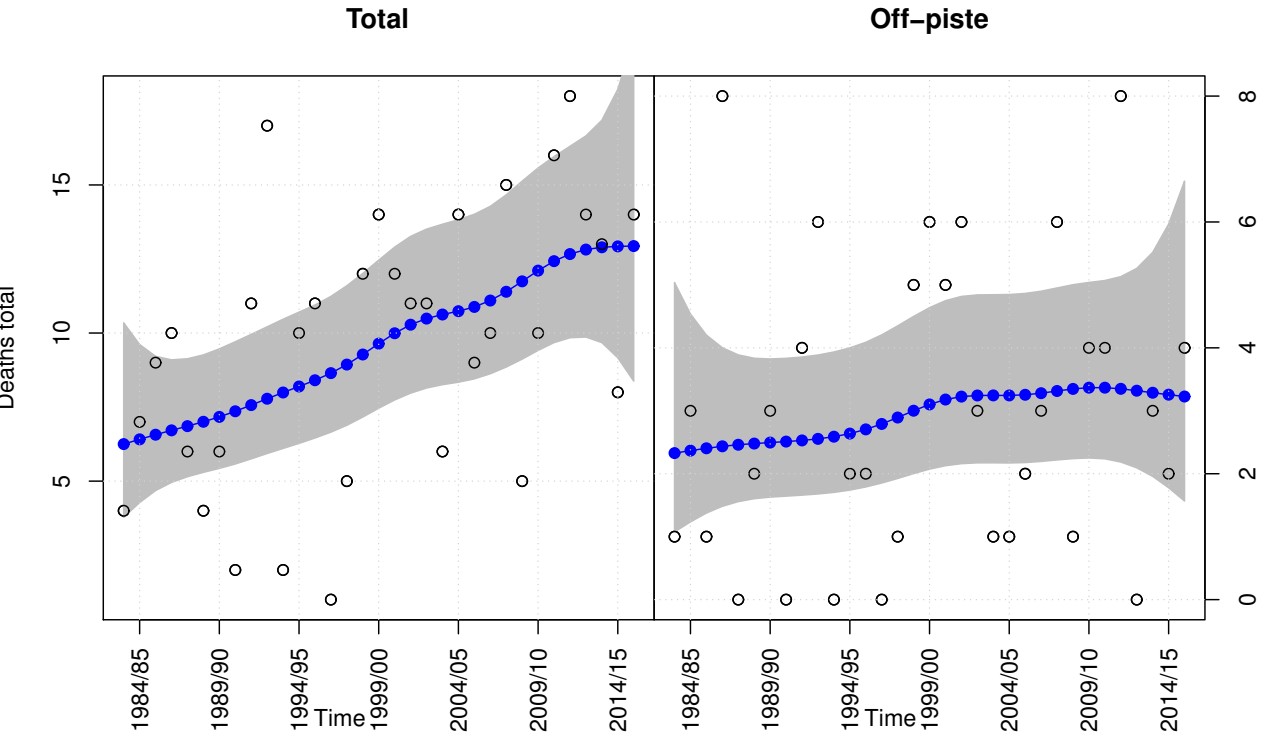

Figure 6: Observed (○) and estimated (●) annual avalanche fatalities (off-piste and backcountry, i.e. total, on the left and off-piste on the right) with 90% confidence bands (grey) in the United States within 1983/84–

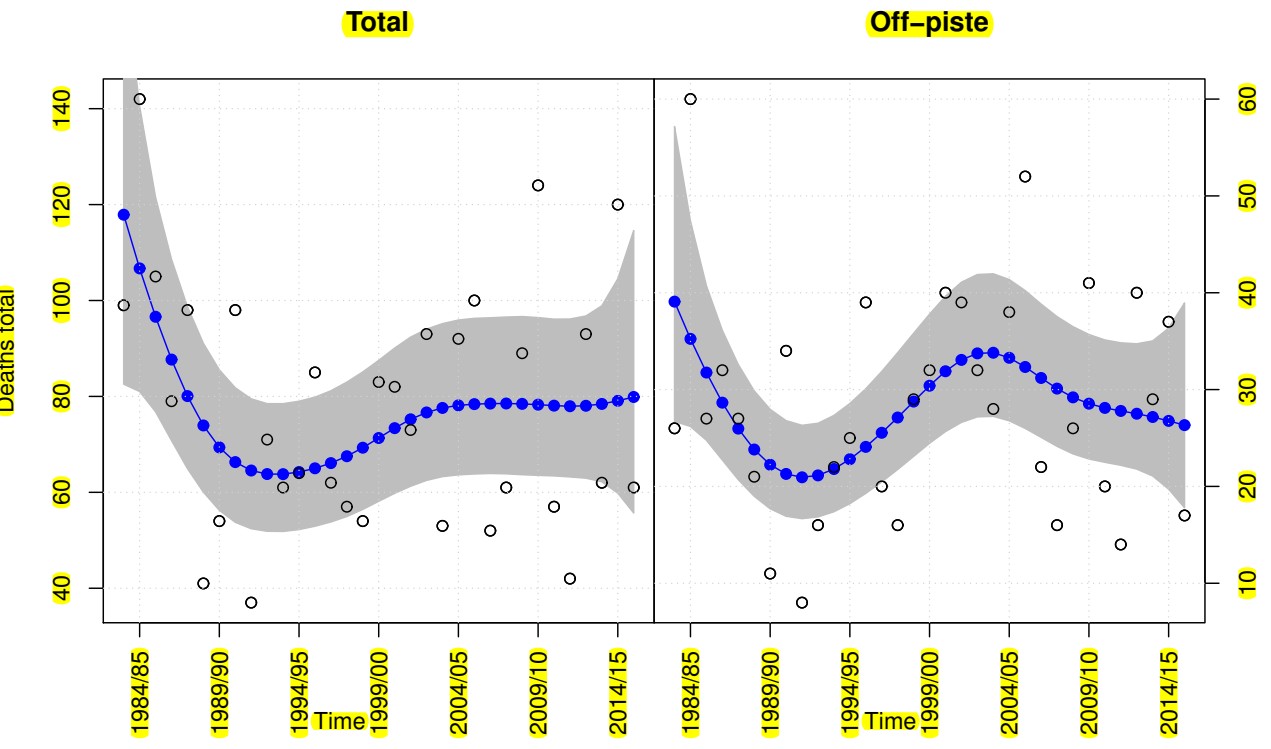

Figure 7: Observed (○) and estimated (●) annual avalanche fatalities (off-piste and backcountry, i.e. total, on the left and off-piste on the right) with 90% confidence bands (grey) in AUT, CHE, FRA and ITA

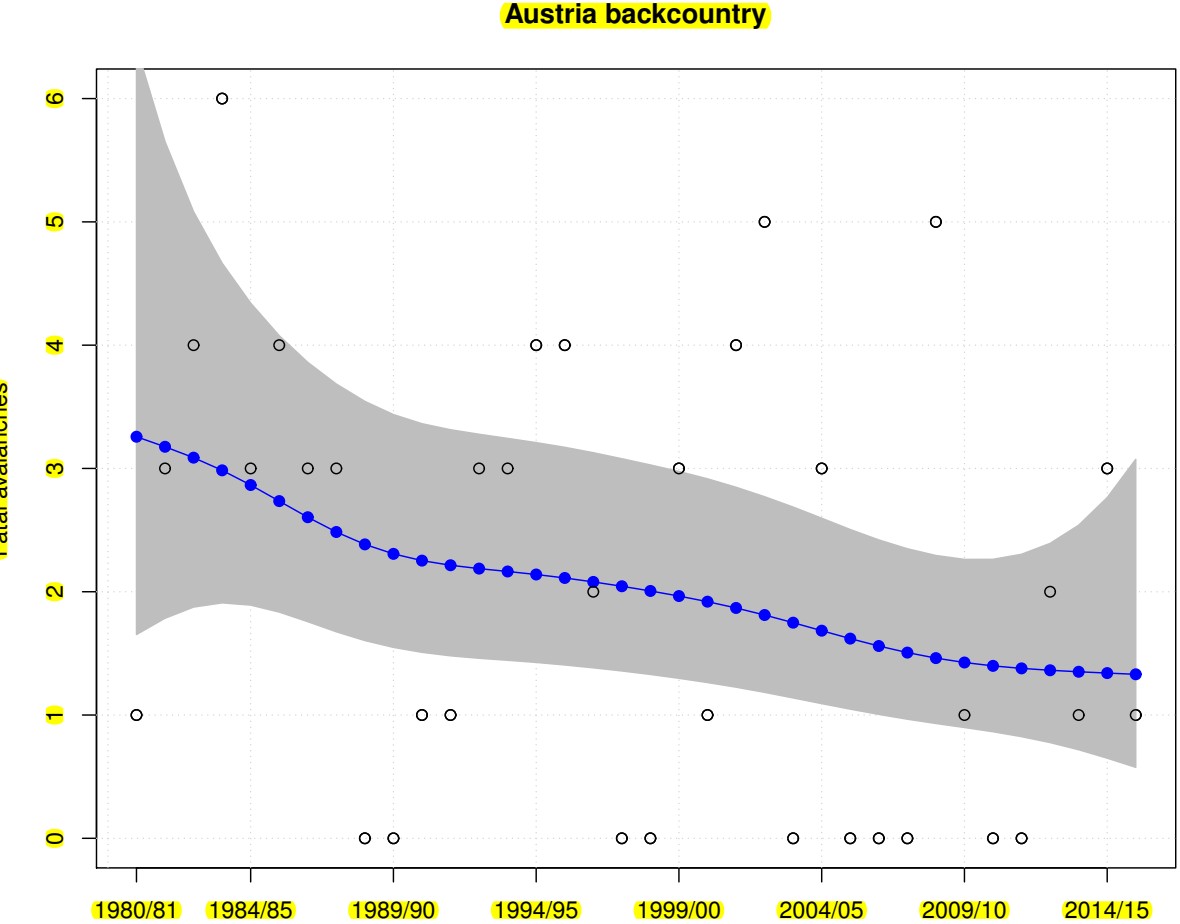

Figure 8: Observed (○) and estimated (●) annual backcountry avalanches (more than 1 fatality) with 90% confidence band (grey) in Austria within 1980/81–2015/16.

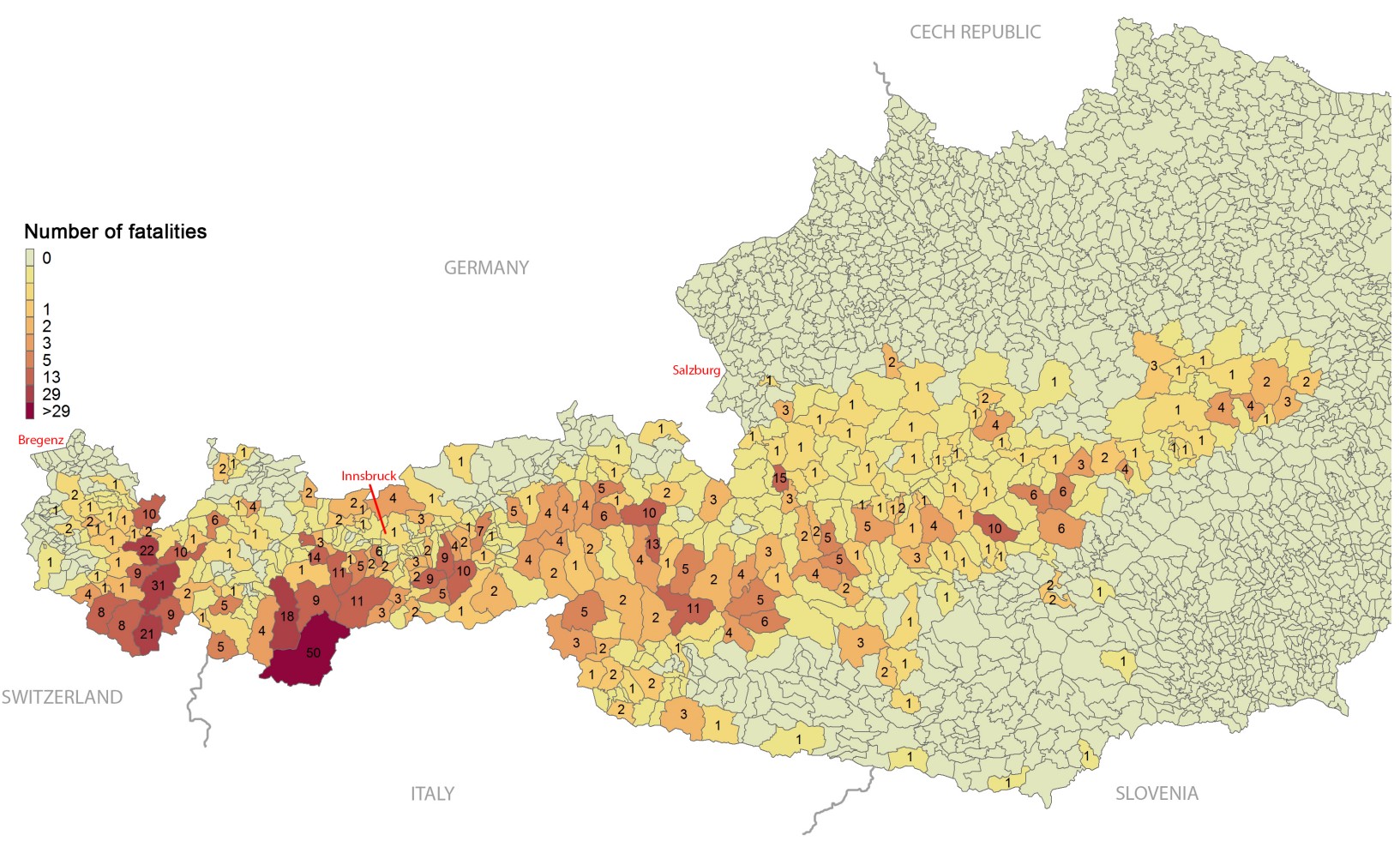

Figure 9: Regional distribution of avalanche fatalities (off-piste and backcountry) in Austria within 1980/81–2015/16.

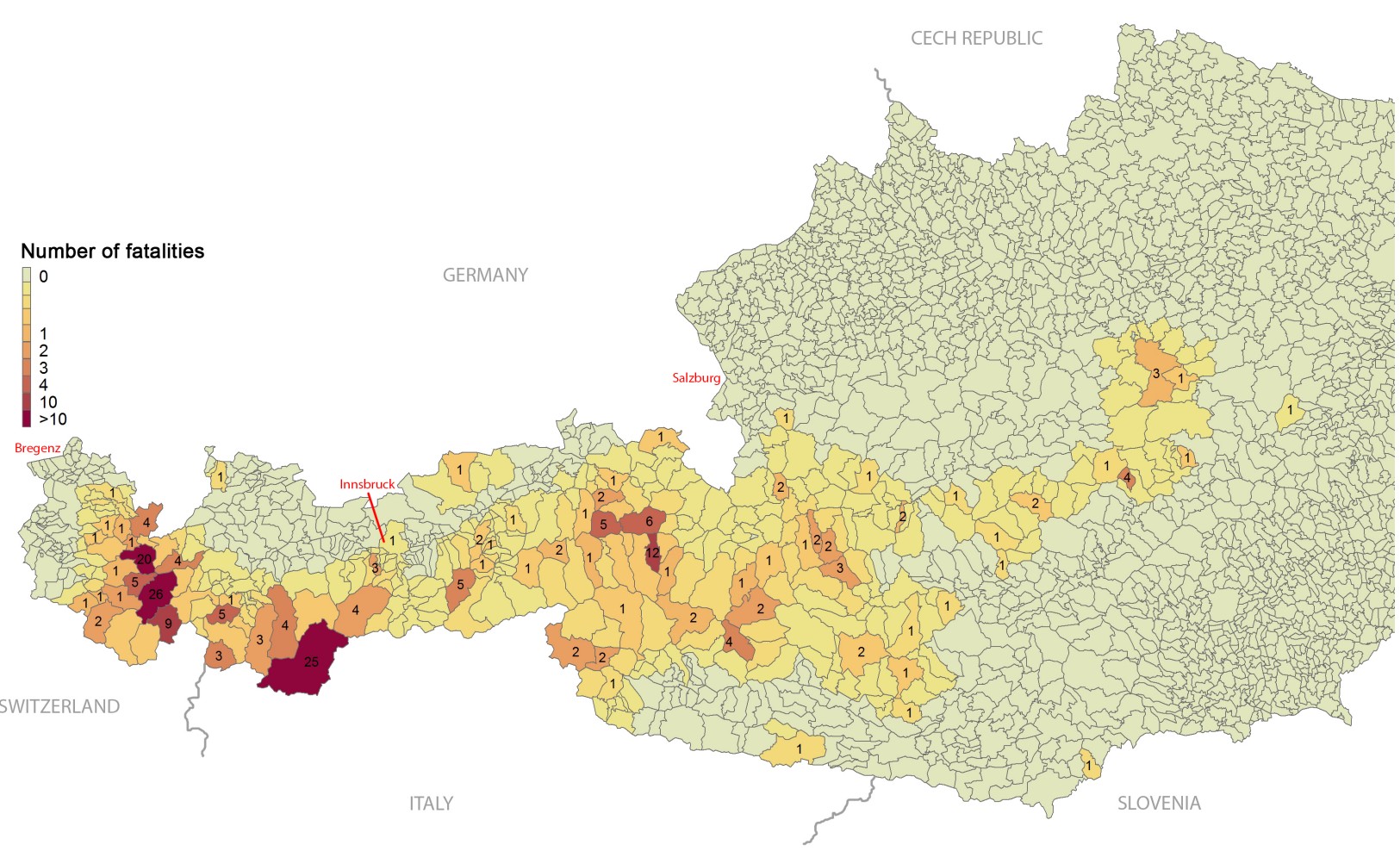

Figure 10: Regional distribution of avalanche fatalities (off-piste) in Austria within 1980/81–2015/16.

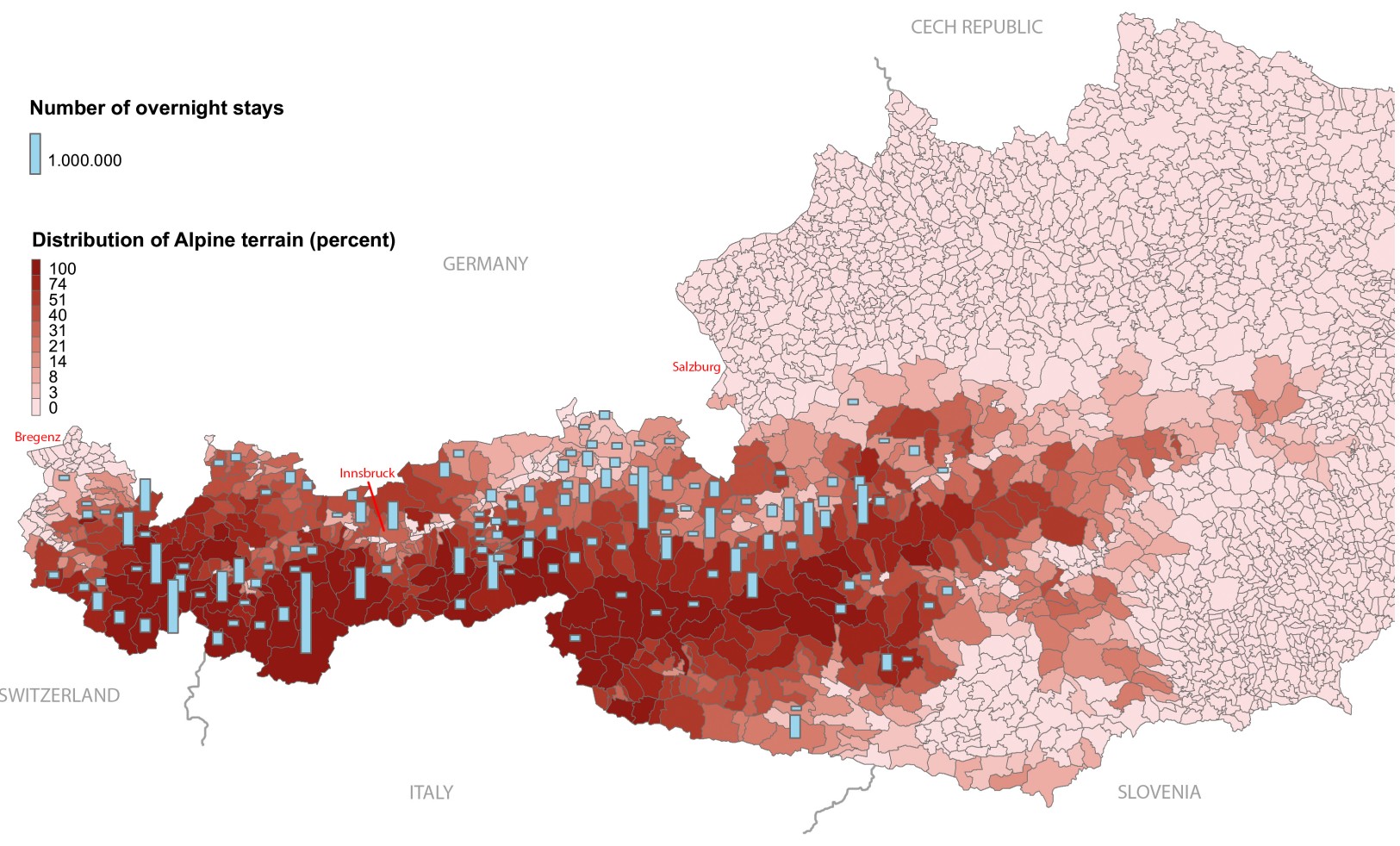

Figure 11: Distribution of Alpine terrain ($\geq 1500m$ above sea level) and number of overnight stays in the winter season 2016 at community level.

| Country | Backcountry | | Off-piste | | Total | | |
|---|---|---|---|---|---|---|---|
| | number | per year | number | per year | number | per year | % off-piste |
| Austria | 458 | 13.88 | 222 | 6.73 | 680 | 20.61 | 32.65% |
| Switzerland | 395 | 11.97 | 222 | 6.73 | 617 | 18.70 | 35.98% |
| France | 433 | 13.12 | 354 | 10.73 | 787 | 23.85 | 44.98% |
| Italy | 322 | 9.76 | 138 | 4.18 | 460 | 13.94 | 30.00% |
| Sum | 1608 | 48.73 | 936 | 28.36 | 2544 | 77.09 | 36.79% |
| USA | 201 | 6.09 | 83 | 2.52 | 284 | 8.61 | 29.23% |

Table 1: Number of avalanche fatalities and annual average (off-piste, back-country and total) of 5 countries within the winter periods 1983/84–2015/16.

| | | Total | | | Off-piste | | |
|---|---|---|---|---|---|---|---|
| | | const | linear | nonlin. | const | linear | nonlin. |
| Austria | AIC | 550.87 | 543.14 | 530.3 | 241.35 | 238.17 | 236.46 |
| | BIC | 552.76 | 546.93 | 539.46 | 243.02 | 241.5 | 243.03 |
| Switzerland | AIC | 256.47 | 254.29 | 242.79 | 189.9 | 191.87 | 186.79 |
| | BIC | 257.96 | 257.29 | 250.1 | 191.4 | 194.87 | 192.83 |
| France | AIC | 268.9 | 270.9 | 267.74 | 251.2 | 253.09 | 245.7 |
| | BIC | 270.39 | 273.89 | 275.39 | 252.7 | 256.08 | 252.28 |
| Italy | AIC | 285.01 | 286.78 | 250.69 | 189.79 | 191.62 | 175.23 |
| | BIC | 286.5 | 289.77 | 257.59 | 191.29 | 194.61 | 180.78 |
| AUT, CHE | AIC | 466.50 | 465.89 | 409.65 | 320.64 | 322.51 | 294.40 |
| FRA and ITA | BIC | 468.00 | 468.88 | 419.14 | 322.13 | 325.50 | 302.29 |
| United States | AIC | 188.64 | 182.43 | 186.33 | 147.45 | 148.92 | 151.33 |
| | BIC | 190.13 | 185.42 | 192.65 | 148.95 | 151.92 | 156.4 |

Table 2: AIC and BIC of the constant, linear and nonlinear trend model considering data of Austria total and off-piste (Figure 1, Figure 2), Switzerland total and off-piste (Figure 3), France total and off-piste (Figure 4), Italy total and off-piste (Figure 5), summing-up of AUT, CHE, FRA, ITA total and off-piste (Figure 7) and United States total and off-piste (Figure 6).

| Community | Backcountry | Off-piste | Total |
|---|---|---|---|
| Sölden | 25 | 25 | 50 |
| St. Anton am Arlberg | 5 | 26 | 31 |
| Lech | 2 | 20 | 22 |
| Galtür | 21 | 0 | 21 |
| St. Leonhard im Pitztal | 14 | 4 | 18 |
| Werfenweng | 13 | 2 | 15 |
| Silz | 14 | 0 | 14 |
| Niedernsill | 1 | 12 | 13 |
| Neustift im Stubaital | 7 | 4 | 11 |
| Heiligenblut am Großglockner | 9 | 2 | 11 |
| St. Sigmund im Sellrain | 11 | 0 | 11 |
| Tux | 5 | 5 | 10 |
| Kaisers | 6 | 4 | 10 |
| Mittelberg | 6 | 4 | 10 |
| Saalbach-Hinterglemm | 4 | 6 | 10 |
| Pusterwald | 9 | 1 | 10 |
| Klösterle | 4 | 5 | 9 |
| Navis | 9 | 0 | 9 |
| Ischgl | 0 | 9 | 9 |
| Längenfeld | 9 | 0 | 9 |
| Wattenberg | 9 | 0 | 9 |
| Gaschurn | 8 | 0 | 8 |
| St. Gallenkirch | 6 | 2 | 8 |
| Fügenberg | 5 | 2 | 7 |
| Jochberg | 1 | 5 | 6 |
| Axams | 3 | 3 | 6 |
| Gaal | 6 | 0 | 6 |
| Häselgehr | 6 | 0 | 6 |
| Wald am Schoberpaß | 6 | 0 | 6 |
| Hohentauern | 4 | 2 | 6 |
| Mallnitz | 6 | 0 | 6 |
| Prägraten am Großvenediger | 5 | 0 | 5 |
| Tweng | 2 | 3 | 5 |
| Nauders | 2 | 3 | 5 |
| Kitzbühel | 3 | 2 | 5 |
| Serfaus | 0 | 5 | 5 |
| Sellrain | 5 | 0 | 5 |
| Schmirn | 5 | 0 | 5 |
| Fusch an der Großglocknerstraße | 5 | 0 | 5 |
| Alpbach | 4 | 1 | 5 |
| Bad Gastein | 3 | 2 | 5 |
| Rohrmoos-Untertal | 5 | 0 | 5 |
| Untertauern | 3 | 2 | 5 |

Table 3: Number of avalanche fatalities (off-piste, backcountry and total) in Austria within 1980/81–2015/16 stratified for communities with more than 4 fatalities in the observation period.

| Date | Location | Municip. | Fatalities |
|---|---|---|---|
| 1982-01-31 | Werfenweng | Werfenweng | 13 |
| 2000-03-28 | Schmiedinger Kogel | Niedernsill | 12 |
| 1999-12-28 | Jamtalhütte - Gde. Galtür | Galtür | 9 |
| 1987-04-05 | Idalpe | Ischgl | 6 |
| 1988-03-28 | Jamtal | Galtür | 6 |
| 2009-05-02 | Schalfkogel | Sölden | 6 |
| 2016-02-06 | Wattener Lizum | Wattenberg | 5 |
| 1985-03-21 | Sonntagkarzinken, Schladm. Tauern | Rohrmoos-Untertal | 4 |
| 1988-02-14 | Hühnereggen, Stubaier Alpen | Sellrain | 4 |
| 1993-04-12 | Querkogeljoch, Ötztaler Alpen | Sölden | 4 |
| 1997-02-18 | Luxnacher Sattel | Häselgehr | 4 |
| 2005-01-22 | Rendl | St. Anton a. Arlberg | 4 |
| 1981-03-01 | Hohe Veitsch | Mürzsteg | 3 |
| 1984-02-19 | Hoher Gleirsch, Karwendelgebirge | Scharnitz | 3 |
| 1985-05-04 | Speikogel, Kitzbüheler Alpen | Westendorf | 3 |
| 1986-01-08 | Kühkarkopf, Hohe Tauern | Fusch a. d. Großglocknerstr. | 3 |
| 1986-04-01 | Tschambreuspitze, Silvretta | Gaschurn | 3 |
| 1986-04-07 | Windachscharte, Stubaier Alpen | Sölden | 3 |
| 1986-12-21 | Lattenberg Triebener Tauern | Wald a. Schoberpaß | 3 |
| 1987-01-06 | Fluchtalpe, Kleines Walsertal | Mittelberg | 3 |
| 1987-04-18 | Scharkogel | Uttendorf | 3 |
| 1991-12-21 | Scharnitzfeld, Wölzer Tauern | Pusterwald | 3 |
| 1995-01-03 | Schöngraben/Törli | St. Anton a. Arlberg | 3 |
| 1995-02-11 | Scheibenspitze | Navis | 3 |
| 1996-03-09 | Frommerkogel, Tennengebirge | Hüttau | 3 |
| 1996-04-03 | Murkarspitze, Gde. Längenfeld | Längenfeld | 3 |
| 2000-03-16 | Wasserradkopf | Heiligenblut | 3 |
| 2000-11-19 | Roßkarschneid | Sölden | 3 |
| 2003-01-30 | Scharnitzalm, Scharnitzfeld | Pusterwald | 3 |
| 2004-12-20 | Mohnenfluh | Lech | 3 |
| 2005-02-22 | Sulzkogel | Silz | 3 |
| 2005-03-05 | Rotschrofenspitze | Kaisers | 3 |
| 2013-01-18 | Mittagskofel, Karnische Alpen | Lesachtal | 3 |

Table 4: List of avalanche events (off-piste or backcountry) in Austria within 1980/81–2015/16 with more than 2 fatalities in each event.