# Peer review of "Spatial and temporal analysis of fatal off-piste and"

_Natural Hazards and Earth System Sciences, 2016_

## Referee Comment (RC1) · Anonymous Referee #1 · 2 Dec 2016

The authors explore trends in the annual number of backcountry avalanche fatalities in Austria and compare these to four other countries. The temporal analysis is carried out applying a generalized additive model. The study evaluates whether linear or non-linear functions describe the annual fatality data best. Additionally, maps showing the spatial distribution of avalanche fatalities by municipality in Austria are presented. These are the novel aspects of the presented work. The topic of the study is within the scope of the journal and will likely be of interest to the journals audience.

[Figure]

**General comments**

I would like to address two main issues concerning the manuscript: (1) the insufficient discussion of the results and embedding of the study within the context of current research, and (2) the time-period analyzed.

(1) Concerning the first point, potentially relevant studies are mentioned in the detailed comments below.

(2) The most recent five years (2011/12 - 2015/16) were not considered in this analysis. However, their inclusion would greatly increase the currency of the analysis. This seems particularly important, as the authors suggest avalanche prevention measures in their study (abstract and lines 282-286). Extending the data-set until 2015/16 would allow a comparison to results shown in recent publications, in which (not significantly) increasing backcountry avalanche fatality numbers were noted during the most recent years (e.g. United States (Birkeland, 2016) and European Alps (Techel et al., 2016)). Therefore, I strongly recommend to include these years, not just for Austria, but also for the other countries.

**Detailed comments, by section**

**Abstract**

l. 29: The study addresses *backcountry* avalanche fatalities, not avalanche fatalities as written.

l. 30-31: There are numerous studies which showed that the backcountry and out-of-bounds avalanche fatality numbers are not constant (e.g. France (Jarry, 2011, Fig. 3); Switzerland (Harvey and Zweifel, 2008); United States (Page et al., 1999); Italy (Valt and Pivot, 2013); European Alps, France, Austria, Switzerland, Italy: (Techel et al., 2016)).

**Introduction**

l. 43-44: specify the "various" reasons which are of special public interest.

l. 47-48: additionally to Brugger et al. (2001), more recent publications should be investigated whether this statement is still considered true (see also the before-mentioned references concerning the abstract)

l. 52: not clear how the citation of Ammann, 2001 is related to the statement by Harvey and Zweifel (2008)

l. 53-55: additionally, in their annual reports the Österreichische Lawinenwarndienste (2016) provide a 20-year overview of the avalanche fatalities in Austria (e.g. Fig. 4, p. 33 in the 2016 report)

l. 66-69: there are brief summaries showing long-term trends of Austrian backcountry fatality statistics in the book by Höller (p. 91, 2015) and also in the 2016 report of the Österreichische Lawinenwarndienste (2016) (pages 210 and 211, results based on Techel et al. (2016))

**Data and methods**

l. 105-111: It should be mentioned, when and how the ICAR data was accessed (URL or citation). It is unclear which of the mentioned ICAR fatality categories were used in the analysis.

l. 117-118 and l. 129-130: I find this very difficult to understand. Did you calculate the trend for each municipality (aggregating the data in terms of location, l. 117-118) separately and then aggregate it again for the regional analysis? Or did you use the annual fatality numbers (all of Austria) for the trend analysis, and the total number of fatalities for each municipality? Please explain this more clearly.

l. 125: You state that in your "opinion" AIC and BIC are better criterion than reporting p-values. You should explain why using AIC and BIC would be more appropriate (advantages, disadvantages). Possibly, you could also give a reference.

[Figure]

**Results and Discussion**

The results section refers to the graphs and tables, but does not present any data. Data is presented mostly in the discussion section. The results should be discussed in more depth than is currently the case.

Trend analysis

The advantage and disadvantage of the proposed statistical approach should be discussed, as this is the main methodological novelty compared to previous publications exploring avalanche fatality statistics. In particular, the following points might be of interest to the reader:

- To what extent do single (or a cluster of) winters with many (or very few) fatalities influence the trend lines shown?

- In your analysis, you analyze subgroups of the data (e.g. off-piste fatalities only). One of the arguments Techel et al. (2016) considered relevant for combining national fatality statistics was the assumption that single multi-fatality events and/or years with many fatalities potentially could have a large effect on trend statistics. Please discuss to what extent this may be relevant, in particular for the trend calculation of the off-piste subgroup, which are characterized by even fewer incidents per year. Please explain whether relatively small accident numbers could be a reason for the sometimes highly fluctuating trend lines (you already briefly comment on this for the Austrian data on lines 193-194).

- The 90% confidence intervals shown in the figures is large at the beginning/end of the time-series. This highlights the greater uncertainty of the trend line calculation. Readers not familiar with confidence intervals, might miss this point when looking at the figures. Therefore, I propose to discuss these uncertainties in the

text.

- Often, the 90% confidence band is relatively wide, which raises the question whether the reported trends can be interpreted as statistically significant. For instance, the trend line of the Swiss off-piste fatalities drops in the nineties and rises in the 2000's. However, the max of the confidence interval in the 1990's is about as high as the minimum in 2000. Therefore, I wonder if the peak around the year 2000 can be considered statistically significant. I recommend you show which of the trends are statistically significant.

You show in Table 2 that the non-linear model is preferable for all the European countries (except for Austrian off-piste fatalities). This is a main result of the study. However, I suggest you discuss potential reasons for the Austrian off-piste fatality trend line being linear, when all the other European trend lines are non-linear.

The trend line for the Swiss backcountry fatalities (Fig. 3) drops from almost 30 in 1983/84 to approximately 15 in the mid-1990's (Fig. 3). This seems like a very strong decrease and is in contrast to the slight but not significant decrease shown/described for the 1990's (e.g. Fig. 3 in SLF (2016) or in Techel et al. (2016)).

On lines 186 to 194 you note a peak in the fatality numbers for Austria in the 1980's, and conclude that higher precipitation during these years might explain this. Looking at off-piste fatalities only, you do not note this peak for Austria. These two statements seem contradictory. It may also be of interest that several authors noted increased numbers of recreational avalanche fatalities in years with less snow (e.g. Luzian, 2000; Valt et al., 2009; Valt and Cianfarra, 2012).

Regional analysis
The regional analysis showed spatial clusters in two regions (Arlberg and Sölden, Fig.

7 and 8). However, an in-depth discussion of potential reasons for these hot-spots is lacking. For instance, visually comparing the clusters shown in Fig. 8 to the size and spatial distribution of ski resorts in Austria (map in Fig. 1 and list of top 20 winter sport municipalities in Fleischhacker (2016)), seems to indicate that these clusters correlate to the spatial distribution of ski resorts in Austria (and hence a greater number of recreationists riding off-piste?). Even though Fuchs et al. (2015) explored the spatial distribution of houses and residents exposed to snow avalanches, the spatial pattern looks again similar to those in Fig.s 7 and 8. with the highest density in the Arlberg and southern Tirol regions.

In general, I would consider it benefitial if you could include other relevant parameters in the spatial analysis. For instance, the spatial clusters of off-piste fatalities could be compared to the distribution and size of the ski areas in the municipalities in Austria (e.g. the data behind the map in Fleischhacker (2016)), while calculating the density of fatalities per surface area above a critical elevation might show if these clusters are related to Alpine topography (e.g. in a Swiss study Techel et al. (2015) considered the elevation range where more than 95% of the recreational accidents occurred).

In the methods section (lines 143-147) you describe the use of Markov random fields to identify the regional hot spots. In the results section and Fig. 7 and 8, it remains unclear how this method was used and what results were obtained. Please highlight what results were gained using this method.

On lines 262-264 you state that you cannot compare spatial patterns to other countries due to lack of information. However, at least for some countries or regions, spatial patterns have been explored and explanations for clusters were given. Relevant publications might include Spencer and Ashley (2010, for the western United States), Logan and Witmer (2012, for Colorado) or Techel et al. (2015, for Switzerland). While Spencer and Ashley argued that these clusters are the areas with the highest concentration of winter sport activities, Logan and Witmer showed that most accidents occurred in areas which are highly accessible (closeness to roads). Techel et al.

concluded that a higher risk to be involved in a backcountry avalanche accident was also correlated to regions with a more frequent shallow snowpack and persistent weak layers. These were not always the regions with the highest number of fatalities.

**Conclusion**
l. 287-297: It is indeed difficult to verify the influence of increased numbers of recreational activity in winter backcountry. The study by Fleischhacker (2016) might provide a suitable reference indicating trends observed in Austrian winter sport regions. A recent study by Winkler (2016, in German) or Winkler et al. (2016, in English) has explored the trends in the number of winter backcountry users in Switzerland during the last two decades. Potentially, this study may be of interest when discussing backcountry usage trends.

**Figures**

Fig. 1 and 2:
The caption should mention that a 90%-confidence interval is shown. Grid lines would be helpful.

Fig. 3 to 6:
The x-axis labeling of the right plot (off-piste) is difficult to read. Maybe leave some space between the plots.
The caption should mention that a 90%-confidence interval is shown.
Grid lines would be helpful.
All these figures, and possibly also the Austrian data for the years 1983/84 until 2010/11 could be presented in a panel plot with the same axis-limits for all countries.
This would facilitate the comparison between the different time-series.

Fig. 7 and 8:
The color choice is difficult to read for colorblind readers. I suggest using any of the color schemes proposed e.g. by Brewer (1994); Neuwirth (2014); Zeileis et al. (2009). Because most readers will be unfamiliar with the Austrian Alps, a map showing the mountainous areas relevant for avalanching - for instance the surface area above 1500 m - would be helpful for comparison.

**References**

Birkeland, K.: U.S. avalanche fatality trend is flat for the past 22 seasons, http://www.fsavalanche.org/news/2016/6/27/ us-avalanche-fatality-trend-is-flat-for-the-past-22-seasons, last access 01.12.2016, 2016.

Brewer, C.: Color use guidelines for mapping, Visualization in modern cartography, pp. 123–148, 1994.

Brugger, H., Durrer, B., Adler-Kastner, L., Falk, M., and Tschirky, F.: Field management of avalanche victims, Resuscitation, 51, 7–15, doi:10.1016/S0300-9572(01)00383-5, 2001.

Fleischhacker, V.: Aktuelle Nachfragetrends im Wintersporttourismus in Österreich, Tech. rep., Institut für touristische Raumplanung ITR, Tulln an der Donau, http://netoffice.salzburgerland. com/downloads/salzburgerland_aktuell/0916%20Studie-Trends-im-Witnersporttourismus. pdf, last access 01.12.2016, 2016.

Fuchs, S., Keiler, M., and Zischg, A.: A spatiotemporal multi-hazard exposure assessment based on property data, Natural Hazards and Earth System Sciences, 15, 2127–2142, doi: 10.5194/nhess-15-2127-2015, http://www.nat-hazards-earth-syst-sci.net/15/2127/2015/, 2015.

Harvey, S. and Zweifel, B.: New trends of recreational avalanche accidents in Switzerland, in: Proceedings International Snow Science Workshop 2008, Whistler, Canada, pp. 9–15, 2008.

Höller, P.: Lawinen - die grössten Katastrophen in Österreich seit 1946/47, Studia Universitätsverlag Innsbruck, 95p., 2015.

Jarry, F.: 40 ans d'accidents d'avalanche . . . 40 ans de prévention, Neige et Avalanches, 135, 18–22, 2011.

Logan, S. and Witmer, F.: Spatial, temporal, and space-time analysis of fatal avalanche accidents in Colorado and the United States, 1991 to 2011, in: Proceedings International Snow Science Workshop, Anchorage, Alaska, USA, pp. 479–486, http://arc.lib.montana.edu/snow-science/objects/issw-2012-479-486.pdf, 2012.

Luzian, R.: Lawinenschäden in Österreich in der Periode von 1967/68 bis 1992/93, in: Proceedings Interpraevent 2000, Villach, Austria, pp. 437–450, 2000.

Neuwirth, E.: RColorBrewer: ColorBrewer Palettes, https://CRAN.R-project.org/package=RColorBrewer, r package version 1.1-2, 2014.

Österreichische Lawinenwarndienste: Winterberichte, https://lawine.tirol.gv.at/archiv/winterberichte/, last access: 01.12.2016, 2016.

Page, C., Atkins, D., Shockley, L., and Yaron, M.: Avalanche deaths in the United States: a 45-year analysis, Wilderness Environ Med., 10, 146–151, 1999.

SLF: Lawinen in der Schweiz: Entwicklung in den letzten 80 Jahren, http://www.slf.ch/praevention/lawinenunfaelle/Publikationen/lawinenunfaelle/Spezialthemen/unfaelle_80jahre/index_DE, last access: 01.12.2016, 2016.

Spencer, J. and Ashley, W.: Avalanche fatalities in the western United States: a comparison of three databases, Natural Hazards, doi:10.1007/s11069-010-9641-3, 2010.

Techel, F., Zweifel, B., and Winkler, K.: Analysis of avalanche risk factors in backcountry terrain based on usage frequency and accident data in Switzerland, Nat. Hazards Earth Syst. Sci., 15, 1985–1997, doi:10.5194/nhess-15-1985-2015, 2015.

Techel, F., Jarry, F., Kronthaler, G., Mitterer, S., Nairz, P., Pavšek, M., Valt, M., and Darms, G.: Avalanche fatalities in the European Alps: long-term trends and statistics, Geographica Helvetica, 71, 147–159, doi:10.5194/gh-71-147-2016, http://www.geogr-helv.net/71/147/2016/, 2016.

Valt, M. and Cianfarra, P.: 1981-2010 avalanche accidents in Italy, in: 4th International Conference on Avalanches and related subjects , September 2011, Kirovsk, Russia, pp. 94–98, 2012.

Valt, M. and Pivot, S.: Avalanche accident documentation is of fundamental importance to understand the dynamics, taking place in snow, of risky activities in order to implement the best possible prevention strategies, in: Proceedings International Snow Science Workshop 2013, Grenoble-Chamonix Mont Blanc, France, pp. 1142–1147, 2013.

Valt, M., Chiambretti, I., and Zasso, R.: 1985 - 2009 twenty-five years of avalanche accidents in Italy, in: Proceedings International Snow Science Workshop, Davos, Switzerland, pp. 686–690, http://www.arpa.veneto.it/temi-ambientali/neve/file-e-allegati-1/05_mv_ISSW09_paper_198.pdf, 2009.

Winkler, K.: Entwicklung des Lawinenrisikos bei Aktivitäten im freien Gelände, alpinforum, pp. 26–33, http://www.slf.ch/dienstleistungen/news/entwicklung_anzahl_lawinenopfer/Jb2016_Final_PDF_Kurt_Winkler.pdf, 2016.

Winkler, K., Fischer, A., and Techel, F.: Avalanche risk in winter backcountry touring: status and recent trends in Switzerland, in: Proceedings International Snow Science Workshop, Breckenridge, Co., pp. 270–276, 2016.

Zeileis, A., Hornik, K., and Murrell, P.: Escaping RGBland: Selecting Colors for Statistical Graphics, Computational Statistics & Data Analysis, 53, 3259–3270, doi:10.1016/j.csda.2008.11.033, https://eeecon.uibk.ac.at/~zeileis/papers/Zeileis+Hornik+Murrell-2009.pdf, 2009.

---

## Short Comment (SC1) · 7 Dec 2016

Dear authors

we – the maintainers of the accident databases in Switzerland and Italy - have noted some differences between the fatality numbers recorded in our databases and the ones shown in Table 1 in your manuscript (see table below this comment).

As it did not become clear, how you defined 'winter' or which of the ICAR categories were considered for this analysis, we recommend to clarify this. You should also

consider that ICAR is potentially not the best datasource for such an analysis, as the national databases are likely more accurate and up-to-date.

When revising the manuscript, we would consider it valuable if you could include the most recent years. Most of these data can be accessed on the internet from the respective national organisations, e.g.:

- Switzerland / SLF: http://www.slf.ch/praevention/lawinenunfaelle/unfaelle_langj/index_DE

- United States / CAIC: http://avalanche.state.co.us/accidents/statistics-and-reporting/

- France / ANENA: http://www.anena.org/5041-bilan-des-accidents.htm

In the case of the Swiss and Italian data, we provide data.

Frank Techel and Benjamin Zweifel, WSL Institute for Snow and Avalanche Research SLF Davos, Switzerland; techel@slf.ch, zweifel@slf.ch
Mauro Valt, Centro Valanghe di Arrabba, Italy - mauro.valt@arpa.veneto.it

[Figure]

Interactive
comment

**Table 1.** Comparison between fatalities as shown in the manuscript and as recorded in the Swiss and Italian avalanche accident databases for the years from 1983/84 until 2010/11. The first number is the one recorded in the national database, the second number the one shown in the manuscript.

| country | backcountry | off-piste |
|---|---|---|
| Italy | 262 vs. 264 | 108 vs. 100 |
| Switzerland | 344[1] vs. 335 | 191 vs. 181 |

[1] 344 includes all backcountry fatalities recreating on skis, snowboards or with snowshoes at the time of avalanche occurrence, additionally there were 85 other backcountry victims during these years (many of these would probably not considered 'winter' avalanches).

**Fig. 1.**

---

## Referee Comment (RC2) · Anonymous Referee #2 · 20 Mar 2017

For clarification: I was asked to do this review about 3 months after the first reviewer finished his/her review. RC1 is very detailed, and I strongly agree with reviewer 1, so I will just add some comments that I find worth to add:

The authors explore trends in the annual number of backcountry avalanche fatalities in Austria and compare these to four other countries. 2 types of studies were executed. While the temporal analysis has some new findings and seems interesting for publication (when the concerns of reviewer 1 are addressed) the regional, spatial analysis is in my opinion not acceptable for publication (I would just skip that part). As reviewer 1

already mentioned, the spatial analysis lacks of correlation to actual skier/snowboarder frequency data, the maps (figure 7 and 8) are misleading in the current form, as the just represent where in Austria popular ski and free ride resorts are, but have no meaning if the chance is actual higher to have an avalanche accident in this particular regions (what the authors claim). If we just look at the 2 hot spots found (Arlberg and southern Ötztal) snow pack conditions are very different. While in Sölden, for example, an inner-alpine snow pack allows for rather dangerous avalanche conditions (shallow cold high altitude snow packs), the Arlberg has often completely different snow pack conditions (warm, heavy snow fall at the border of the Alps with lower altitude). At the Arlberg the huge amount of skiers going off-piste and back country skiing rather explain the frequency of avalanche accidents. I am completely aware that skiers/snowboarder frequency data is difficult to get in a meaningful way (reviewer 1 had some good ideas). I could also suggest using data of ski-tickets sold per day (available from the ski resorts) or statistics of guest-nights (overnight statistics available at the Austrian chamber for tourism) but I think it will be still very difficult to create a meaningful map, so as mentioned I would skip the regional analysis.

In the temporal analysis I would add at least in the discussion that the number of skiers/snowboarders or winter tourists increased in the period investigated (for example in Tirol winter guests increased from 1986 being 2.922.842 to 2016 being 5.819.984 https://www.tirol.gv.at/statistik-budget/statistik/tourismus/) or use alternative statistics. That fact needs to be discussed in more detail (as reviewer 1 already mentioned) as clearly a boom in back country skiing and off-piste skiing has happened in the last decades. So even if you see a slightly increasing trend of fatalities in Austria it is definitely not an increasing trend when we account for skier/snowboarder frequency.

---

## Author Response (AR1)

The authors explore trends in the annual number of backcountry avalanche fatalities in Austria and compare these to four other countries. The temporal analysis is carried out applying a generalized additive model. The study evaluates whether linear or non-linear functions describe the annual fatality data best. Additionally, maps showing the spatial distribution of avalanche fatalities by municipality in Austria are presented. These are the novel aspects of the presented work. The topic of the study is within the scope of the journal and will likely be of interest to the journals audience.

**General comments**

I would like to address two main issues concerning the manuscript: (1) the insufficient discussion of the results and embedding of the study within the context of current research, and (2) the time-period analyzed.

(1) Concerning the first point, potentially relevant studies are mentioned in the detailed comments below.

**(Our point by point reply to the reviewer's comments in bold)**
**Thank you for the many constructive comments. We will bring up to date the discussion of our results in the context of the relevant literature. See below for more comments and details regarding the suggested references.**

(2) The most recent five years (2011/12 - 2015/16) were not considered in this analysis. However, their inclusion would greatly increase the currency of the analysis. This seems particularly important, as the authors suggest avalanche prevention measures in their study (abstract and lines 282-286). Extending the data-set until 2015/16 would allow a comparison to results shown in recent publications, in which (not significantly) increasing backcountry avalanche fatality numbers were noted during the most recent years (e.g. United States (Birkeland, 2016) and European Alps (Techel et al., 2016)). Therefore, I strongly recommend to include these years, not just for Austria, but also for the other countries.

**For the new version of the article we extended the database (for both: Austria and other countries for comparison) up to the winter period 2015/16 using national data in case of Austria and ICAR**

**data in case of the other countries. Further on we checked the data according to the comments of Techel in the SC1 (Swiss Data: Auszug der Lawinendatenbank des SLF; Italian data: Mauro Valt: Associazione Interregionale Neve e Valanghe, Trento). Additionally, a similar crosscheck was made with the French and the US data (Frederic Jarry: ANENA; Ethan Greene: Colorado Avalanche Information Center).**

**(Originally, the database of the survey was established in 2011 within the frame of a research seminar at the University of Innsbruck).**

**Detailed comments, by section**

**Abstract**
l. 29: The study addresses backcountry avalanche fatalities, not avalanche fatalities as written.

**We changed to "backcountry and off-piste avalanche fatalities ...", see line 29.**

l.30-31: There are numerous studies which showed that the backcountry and out-of-bounds avalanche fatality numbers are not constant (e.g. France (Jarry, 2011, Fig. 3); Switzerland (Harvey and Zweifel, 2008); United States (Page et al., 1999); Italy (Valt and Pivot, 2013); European Alps, France, Austria, Switzerland, Italy: (Techel et al., 2016)).

**We mean relating to Austrian data. We changed to:**

**"to the widespread opinion in Austria, that the number....", see line 31**

**Here are some comments (press, World Wide Web, literature) referring to Austria:**

**derStandard 15.1.2012: 25 Lawinentote werden akzeptiert citing Thomas Wiesinger (Universität für Bodenkultur); Lawinenkolloquium 2012 Salzburg:**

**"Je nach Schätzung gibt es in Österreich 350.000 bis 650.000 aktive Skitourengeher. Trotzdem ist die Zahl der Lawinentoten über Jahrzehnte hinweg konstant."**

**Url:**

**http://derstandard.at/1326502791533/Maengel-bei-Lawinenwarnung-25-Lawinentote-werden-akzeptiert**

**SpringerMedizin.at 18.1.2016: Schneemenschen unter sich:**

**"20 Menschen sterben in Österreich jeden Winter den Weißen Tod, sie enden jämmerlich begraben unter Schneebrettern. Doch ihre Zahl bleibt konstant, während sich jene der Skitouren- und Variantengeher der Millionengrenze annähert."**

**Url:**

**http://www.springermedizin.at/schwerpunkt/lebensstil/?full=51211**

**Further on, the book of Elke Roth**

**Roth 2013: Lawinen: verstehen -vermeiden-Praxistipps. Bergverlag Rother, München p141:**

**"Alle Ursachen zusammen haben dazu geführt, dass die Zahl der Lawinentoten in etwa konstant geblieben und nicht mit der Zahl der exponierten Personen gewachsen ist."**

**Citation:**

```
@Book{,
 author = {Roth E.},
 title = { Lawinen: verstehen -vermeiden-Praxistipps },
 year = {2013},
 pages = {303},
 publisher = {Bergverlag Rother},
 address = {München}
```

**Introduction**

l. 43-44: specify the "various" reasons which are of special public interest.

- mass media; bad news are good (interesting) news
- see e.g. public interest in the Galtür 1999 disaster (or in the Eiger north face climbing disaster in 1936)
- public interest of protection against natural hazards

**But we added a citation of the a master thesis from the 1980s in line 45 which addresses this topic:**

```
@Book{,
 author = {Januskovecz A.},
 title = { Zeitungsberichterstattung über Naturkatastrophen, Ansätze für die forstliche
Öffentlichkeitsarbeit zum Thema Lawinen –Hochwasser –Muren },
 year = {1989},
 pages = {112},
 publisher = {Hochschulschrift: Univ. für Bodenkultur, Dipl.-Arb.},
 address = {Wien}
}
```

l. 47-48: additionally to Brugger et al. (2001), more recent publications should be investigated whether this statement is still considered true (see also the before mentioned references concerning the abstract)

**Please see lines 46-57. But note that e.g.**
**in case of France the trend functions of Jarry (2011) indicate rather no positive or negative trend (if anything, the lower counts in the mid 1990's).**

**in case of Italy Valt & Pivot only observed an increase/decrease of the percentage of casualties among backcountry/off-piste skiers.**

l. 52: not clear how the citation of Ammann, 2001 is related to the statement by Harvey and Zweifel (2008)

**You are right, we skipped this citation!**

l. 53-55: additionally, in their annual reports the Österreichische Lawinenwarndienste (2016) provide a 20-year overview of the avalanche fatalities in Austria (e.g. Fig. 4, p. 33 in the 2016 report)

**This is just a copy of the idea of the Kuratorium für alpine Sicherheit (Kurasi) which was used in the recent reports of the ÖLWD. This kind of the graphics is a "tradition" of the Kurasi report since the early 1990s.**

l. 66-69: there are brief summaries showing long-term trends of Austrian backcountry fatality statistics in the book by Höller (p. 91, 2015) and also in the 2016 report of the Österreichische Lawinenwarndienste (2016) (pages 210 and 211, results based on Techel et al. (2016))

**The citation of Höller is referring to a presentation of mine in Palermo 2013 (based on the data of this paper which has not been published yet in a peer reviewed journal)**

**But, thank you for bringing the highly relevant paper of Techel et at. (2016) to our attention. At the time of writing this was not turned up by searches in the Web of Science. Indeed, there are many parallels between our work and that of Techel. However, there are also important differences in the population considered. Specifically, the group of backcountry and off-piste fatalities in our study is just a subset of avalanche fatalities as analyzed in Techel et al.**

**We, of course, will update the discussion considering the new results of Techel et al.**

**Please, see lines 53-57, 266-67 in the new version of the paper.**

**Data and methods**
l. 105-111: It should be mentioned, when and how the ICAR data was accessed (URL or citation). It is unclear which of the mentioned ICAR fatality categories were used in the analysis.

**Due to personal contact of Mr. Höller with the ICAR. We mentioned the ICAR fatality categories in line 110 (which are the lines 113-115 in the new version).**

**And as written above, we will check the ICAR data according the statement of Techel in SC1: "comment on table 1".**

l. 117-118 and l. 129-130: I find this very difficult to understand. Did you calculate the trend for each municipality (aggregating the data in terms of location, l. 117-118) separately and then aggregate it again for the regional analysis? Or did you use the annual fatality numbers (all of Austria) for the trend analysis, and the total number of fatalities for each municipality? Please explain this more clearly.

The meaning is as follows: Aggregating the spatio-temporal DATABASE in terms of municipalities which means summing up all over Austria resulting in annual fatality numbers (or summing up over the years resulting in data stratifed for municipalities).

We changed this in order to be more clear; see line 133 ff.:

"After aggregating the spatio-temporal data y_st (denoting the observed fatalities at time t and location s) in terms of location, which means summing up over the locations,......"

l. 125: You state that in your "opinion" AIC and BIC are better criterion than reporting p-values. You should explain why using AIC and BIC would be more appropriate (advantages, disadvantages). Possibly, you could also give a reference.

Comparisons with p-values (e.g., from likelihood ratio or Wald tests) always pertain to comparisons of pairs of nested models. When a larger number of models has to be compared this typically leads to (a) many pairwise comparisons, (b) possibly non-nested models, (c) multiplicity of tests. Therefore, in such situations information criteria are often used for model selection rather than significance tests. This is particularly popular in regression analysis (see e.g., Venables & Ripley 2002) and ARIMA modeling for time series (see e.g., Cryer & Chan 2008).

@Book{,
 author = {William N. Venables and Brian D. Ripley},
 title = {Modern Applied Statistics with \proglang{S}},
 edition = {4th},
 year = {2002},
 pages = {495},
 publisher = {Springer-Verlag},
 address = {New York}
}

@Book{,
 author = {Jonathan D. Cryer and Kung-Sik Chan},
 title = {Time Series Analysis With Applications in {R}},
 publisher = {Springer-Verlag},
 address = {New York},
 year = {2008}
}

Please, see citation at line 144.

**Results and Discussion**
The results section refers to the graphs and tables, but does not present any data. Data is presented mostly in the discussion section.

**However, the result section refers to tables and figures at the end of the paper (according to the guidelines). Some journals ask for this kind of manuscript composition we did. But nethertheless, we are open for possible changes.**

The results should be discussed in more depth than is currently the case.

**We would like, if wished, to extend the discussion adding**

- **the tables of avalanche counts for municipalities with the most avalanche events in Austria**
- **the list of avalanche events with highest counts**

**in the regional part of the paper. We skipped these tables in the current version in order to keep the paper short (instead of tables we tried to use citations, see e.g. line 261).**

**In any case, there are further points which we would like to address in the discussion, see below.**

**In the new version, at least, we added the tables as described above. Further on, we extended the discussion in the "temporal" and the "regional" part considerably.**

Trend analysis
The advantage and disadvantage of the proposed statistical approach should be discussed, as this is the main methodological novelty compared to previous publications exploring avalanche fatality statistics. In particular, the following points might be of interest to the reader:

- To what extent do single (or a cluster of) winters with many (or very few) fatalities influence the trend lines shown?

**A good question! One single extreme event (winter) has almost no effect on the nonlinear trend function. In our opinion, the GAM estimator behaves robust for this data (in contrast to the linear model or the running mean of Techel et al. 2016). See e.g. the single extreme winter (>=40) of "Austria total" in the early 1970s or the single extreme winter of "France total" in the early 1990s.**

**There are clusters of winters which do have an influence on the profiles e.g.:**

- **Austria total (6 larger values) in the mid 1980s - see paper line 186-194 (214-219 new version)**
- **Switzerland off-piste (5 smaller values) in the early 1990s**
- **France total (5 smaller values) around 1990; despite the single extreme event mentioned above**
- **Italy total (5 smaller values) in the mid 1990s.**

**We addressed these points (especially the clusters of "extreme" winters) in the final version, see lines 268-276. Thank you for this advice.**

- In your analysis, you analyze subgroups of the data (e.g. off-piste fatalities only). One of the arguments Techel et al. (2016) considered relevant for combining national fatality statistics was the assumption that single multi-fatality events and/or years with many fatalities potentially could have a large effect on trend statistics.

**We do not think that single multi fatality events have an influence on the GAM estimator; see our comment earlier. Single multi-fatality events in Austria, e.g. Werfenweng 1982 (13 fatalities), Niedensill 2000 (12), Galtür 1999 (9), have an influence on the Markov random field (MRF) estimator (see discussion line 260) but not on the estimated temporal profile of Figure 1.**

Please discuss to what extent this may be relevant, in particular for the trend calculation of the off-piste subgroup, which are characterized by even fewer incidents per year. Please explain whether relatively small accident numbers could be a reason for the sometimes highly fluctuating trend lines (you already briefly comment on this for the Austrian data on lines 193-194).

**Because of the smoothness of the GAM estimator, we do not observe fluctuating trend lines (which is the case if we use the running mean, see Techel et al. (2016)), even if the accident numbers are rather small.**

**(Maybe in case of Austrian off-piste data, we assume some uncertainty because of a boundary effect at the end of the temporal profile, see the following:)**

- The 90% confidence intervals shown in the figures is large at the beginning/end of the time-series. This highlights the greater uncertainty of the trend line calculation. Readers not familiar with confidence intervals, might miss this point when looking at the figures. Therefore, I propose to discuss these uncertainties in the text.

**These effects are due to boundary effects which are well known in the analysis of time dependent data. As a result of observing no data on the left at the beginning and no data on the right at the end, the estimates at the beginning and the end are more uncertain.**

**Thank you, we mentioned this point in the discussion of the final version, e.g. see lines 268-276.**

- Often, the 90% confidence band is relatively wide, which raises the question whether the reported trends can be interpreted as statistically significant. For instance, the trend line of the Swiss off-piste fatalities drops in the nineties and rises in the 2000's. However, the max of the confidence interval in the 1990's is about as high as the minimum in 2000. Therefore, I wonder if the peak around the year 2000 can be considered statistically significant. I recommend you show which of the trends are statistically significant.

**Our AIC/BIC approach is model selection between the constant, linear, or nonlinear model on the whole. In this paper we did not test significances for subintervals (eg. >=2000) knowing that the**

number of cases would be too small. We are only able to give descriptive analysis (more or less by visual inspection): In case of Switzerland off-piste, the nonlinear model is preferable*; we notice smaller counts in the early 1990s (please take notice of a cluster with 4 (or 5) small values) and large(r) counts in the early 2000s. See also Techel et al. (2016) with larger number of counts in the early 2000s.

Maybe, the extreme estimates in the early 1980s are due to uncertainties because of the boundary effect as described above.

*please note, that in case of Swiss off-piste fatalities the BIC values based on the new (extended) data almost indicate that the constant model is appropriate.

You show in Table 2 that the non-linear model is preferable for all the European countries (except for Austrian off-piste fatalities). This is a main result of the study. However, I suggest you discuss potential reasons for the Austrian off-piste fatality trend line being linear, when all the other European trend lines are non-linear. The trend line for the Swiss backcountry fatalities (Fig. 3) drops from almost 30 in 1983/84 to approximately 15 in the mid-1990's (Fig. 3). This seems like a very strong decrease and is in contrast to the slight but not significant decrease shown/described for the 1990's (e.g. Fig. 3 in SLF (2016) or in Techel et al. (2016)).

Good question: It could be some uncertainty at the beginning of the time profile (larger confidence band). Another reason could be that the data of Techel 2016 are different to our data ("uncontrolled terrain").

However, we will mention in the final version that the GAM estimates of the early 1980 Swiss-backcountry counts (maybe others too?!) are rather uncertain because of the large confidence band at the beginning – see lines 268-276 of the new version and the discussion above.

On lines 186 to 194 you note a peak in the fatality numbers for Austria in the 1980's, and conclude that higher precipitation during these years might explain this. Looking at off-piste fatalities only, you do not note this peak for Austria. These two statements seem contradictory. It may also be of interest that several authors noted increased numbers of recreational avalanche fatalities in years with less snow (e.g. Luzian, 2000; Valt et al., 2009; Valt and Cianfarra, 2012).

It is supposed that increased snowfall has an effect on increased avalanche counts (although not fully examined and published, we have some evidence for this in our research, e.g. increased snowfall in the 80's in the "St. Anton" cluster).

However, we have no idea (empirical explanation, citations which we could mention in the paper) why there is a peak in the total case and no peak in the off-piste case. We simply observe that increased snowfall in the 1980s has no effect on off-piste avalanche fatalities in Austria. Last but not least, we observe larger counts of off-piste fatalities in the 1980s if we look at the counts of Switzerland, France and Italy.

Regional analysis
The regional analysis showed spatial clusters in two regions (Arlberg and Sölden, Fig. 7 and 8).

However, an in-depth discussion of potential reasons for these hot-spots is lacking. For instance, visually comparing the clusters shown in Fig. 8 to the size and spatial distribution of ski resorts in Austria (map in Fig. 1 and list of top 20 winter sport municipalities in Fleischhacker (2016)), seems to indicate that these clusters correlate to the spatial distribution of ski resorts in Austria (and hence a greater number of recreationists riding off-piste?). Even though Fuchs et al. (2015) explored the spatial distribution of houses and residents exposed to snow avalanches, the spatial pattern looks again similar to those in Fig.s 7 and 8. with the highest density in the Arlberg and southern Tirol regions.

**Thank you for this interesting congruity, we will add these citations for discussion in the final version, see lines, 339-346.**

In general, I would consider it benefitial if you could include other relevant parameters in the spatial analysis. For instance, the spatial clusters of off-piste fatalities could be compared to the distribution and size of the ski areas in the municipalities in Austria (e.g. the data behind the map in Fleischhacker (2016)), while calculating the density of fatalities per surface area above a critical elevation might show if these clusters are related to Alpine topography (e.g. in a Swiss study Techel et al. (2015) considered the elevation range where more than 95% of the recreational accidents occurred).

**We add a map visualizing the municipal Alpine terrain (>=1500m) with additional information (points) of the 50 largest municipalities relating to overnight stays in the winter season 2016. We are able to calculate this using an Austrian digital elevation model and the information of overnight stays from Austria (instead of the federal states Vorarlberg, Tyrol and Salzburg as proposed). As a result of this we are able to compare the maps in the discussion (which I prefer from an epidemiologic point of view instead of calculating the density of fatalities).**

In the methods section (lines 143-147) you describe the use of Markov random fields to identify the regional hot spots. In the results section and Fig. 7 and 8, it remains unclear how this method was used and what results were obtained. Please highlight what results were gained using this method.

**Spatial estimates were calculated with the MRF model and the colorings of the maps are based on these estimates. The spatial estimates were only used for the coloring in order to explore regional clusters with visual inspection. See lines 195-198.**

On lines 262-264 you state that you cannot compare spatial patterns to other countries due to lack of information. However, at least for some countries or regions, spatial patterns have been explored and explanations for clusters were given. Relevant publications might include Spencer and Ashley (2010, for the western United States), Logan and Witmer (2012, for Colorado) or Techel et al. (2015, for Switzerland). While Spencer and Ashley argued that these clusters are the areas with the highest concentration of winter sport activities, Logan and Witmer showed that most accidents occurred in areas which are highly accessible (closeness to roads). Techel et al. concluded that a higher risk to be involved in a backcountry avalanche accident was also correlated to regions with a more frequent shallow snowpack and persistent weak layers. These were not always the regions with the highest number of fatalities.

**We will take this into account in the discussion of the final paper; thank you for this advice.**

**We skipped the lines 262-264 and we added the citation of Spencer and Ashley, see line 340 in the new version. Looking at the proceeding paper of Logan and Witmer I am not shure about the validity of the statement above.**

**However, some issues (shallow snowpack and persistent weak layer) are topic of our research proposal which we submitted a few months ago.**

**Conclusion**
l. 287-297: It is indeed difficult to verify the influence of increased numbers of recreational activity in winter backcountry. The study by Fleischhacker (2016) might provide a suitable reference indicating trends observed in Austrian winter sport regions. A recent study by Winkler (2016, in German) or Winkler et al. (2016, in English) has explored the trends in the number of winter backcountry users in Switzerland during the last two decades. Potentially, this study may be of interest when discussing backcountry usage trends.

**We will take this into consideration for discussion, see lines 331 ff. of the new version. However, one very important part of our submitted research project (spatio-temporal model) is to get reliable information on the number backcountry and off-piste skiers in general.**

**Figures**

Fig. 1 and 2:
The caption should mention that a 90%-confidence interval is shown. Grid lines would be helpful.

**Thank you for this advice.**

Fig. 3 to 6:
The x-axis labeling of the right plot (off-piste) is difficult to read. Maybe leave some space between the plots.

**We did this in order to gain space for the plots, we tried some versions (among them with space between the plots) and decided for the current version. But, we are able to put the axis labels of the second plot to the right side (which is a good solution if we add grid lines).**

The caption should mention that a 90%-confidence interval is shown. Grid lines would be helpful.

**Thank you for this advice, see the grid lines in the new version.**

All these figures, and possibly also the Austrian data for the years 1983/84 until 2010/11 could be presented in a panel plot with the same axis-limits for all countries. This would facilitate the comparison between the different time-series.

**We have some concerns about that because of readability. However, it is possible to "pile up" the plots with the same x-axis (omitting multiple labels) in order to save space.**

Fig. 7 and 8:
The color choice is difficult to read for colorblind readers. I suggest using any of the color schemes proposed e.g. by Brewer (1994); Neuwirth (2014); Zeileis et al. (2009). Because most readers will be unfamiliar with the Austrian Alps, a map showing the mountainous areas relevant for avalanching - for instance the surface area above 1500 m - would be helpful for comparison.

**The colors indicate:**

- **Green: no danger**
- **Red: danger**

**But, we are open for other color schemes when generating the maps with the new data.**

**Please take note of our proposal of map #3 above.**

**→ As a result of further discussion between the authors of the article we changed the color scheme to: dark red-light yellow**

**Anonymous Referee #2**

For clarification: I was asked to do this review about 3 months after the first reviewer finished his/her review. RC1 is very detailed, and I strongly agree with reviewer 1, so I will just add some comments that I find worth to add: The authors explore trends in the annual number of backcountry avalanche fatalities in Austria and compare these to four other countries. 2 types of studies were executed. While the temporal analysis has some new findings and seems interesting for publication (when the concerns of reviewer 1 are addressed) the regional, spatial analysis is in my opinion not acceptable for publication (I would just skip that part).

As reviewer 1 already mentioned, the spatial analysis lacks of correlation to actual skier/snowboarder frequency data, the maps (figure 7 and 8) are misleading in the current form, as the just represent where in Austria popular ski and free ride resorts are, but have no meaning if the chance is actual higher to have an avalanche accident in this particular regions (what the authors claim).

**Please, also take notice that reviewer #1 referred to some citations with spatial results (without any explaining variables).**

If we just look at the 2 hot spots found (Arlberg and southern Ötztal) snow pack conditions are very different. While in Sölden, for example, an inneralpine snow pack allows for rather dangerous avalanche conditions (shallow cold high altitude snow packs), the Arlberg has often completely different snow pack conditions (warm, heavy snow fall at the border of the Alps with lower altitude).

**Do you have a citation for this (relating to different snowpack conditions considering the Arlberg or Sölden)? It would be of some interest for us; as stated above, this is part of our research planned in the future.**

At the Arlberg the huge amount of skiers going off-piste and back country skiing rather explain the frequency of avalanche accidents. I am completely aware that skiers/snowboarder frequency data is difficult to get in a meaningful way (reviewer 1 had some good ideas). I could also suggest using data of ski-tickets sold per day (available from the ski resorts) or statistics of guest-nights (overnight statistics available at the Austrian chamber for tourism) but I think it will be still very difficult to create a meaningful map, so as mentioned I would skip the regional analysis.

**We appreciate that reviewer #2 considers the temporal analysis to be an interesting (in his opinion the only interesting) part of our contribution. However, we feel that there are still interesting insights from the spatial analysis that are worth to be discussed in this publication. As already pointed out in the reply to the reviewer #1, we have tried to improve the spatial analysis, i.e. specifically (a) adding 2 tables for regional discussion, (b) generating map #3 as described above.**

**We think that the spatial analysis is meaningful in terms of prevention if we consider the narrow regional distribution of the fatalities, see conclusion line 281 (now lines 363-ff).**

In the temporal analysis I would add at least in the discussion that the number of skiers/snowboarders or winter tourists increased in the period investigated (for example in Tirol winter guests increased from 1986 being 2.922.842 to 2016 being 5.819.984 https://www.tirol.gv.at/statistik-budget/statistik/tourismus/) or use alternative statistics. That fact needs to be discussed in more detail (as reviewer 1 already mentioned) as clearly a boom in back country skiing and off-piste skiing has happened in the last decades. So even if you see a slightly increasing trend of fatalities in Austria it is definitely not an increasing trend when we account for skier/snowboarder frequency.

**Thank you for this advice, see e.g. lines 372-374. But, if we e.g. compare the temporal profile of winter overnight stays in Tirol since 1986 with those of the avalanche fatalities the congruence is rather weak. It would be of some interest if there is a congruity in case of the off-piste fatality centers Sölden and St. Anton? But so far we have only data for these municipalities beginning at 2000.**

**However, we think that the size of tourist resorts is misleading in case of backcountry skiers (which are more or less native if we consider for example backcountry skiers around Innsbruck).**

[revised manuscript text omitted]

---

## Author Response (AR2)

**Reviewer #1:**

Literature

One of the co-authors (Peter Höller) has recently published an in-depth analysis of Austrian avalanche fatalities, with a particular emphasis on tourist avalanches (Höller, 2017). Höllers study covers essentially the same two categories backcountry and off-piste, and a very similar time period. Currently, Höller's study is not cited. From my perspective, the following results and statements made by Höller should be discussed, as they partly contradict the results and/or may influence the interpretations presented:
– frequency and influence of multi-fatality accidents - 32 accidents with more than 3 victims in the backcountry since 1981/82
– «Although in Austria a slight increase of fatalities in the off-piste area can be seen, this tendency is statistically not significant (p = 0.055).»(p. 6)
– «A trend towards more avalanche fatalities due to off-piste skiing cannot be identified at the moment. »(p. 7)

**Thank you for this advice! I did not know this paper. However, the data which are denoted by "tourist avalanches" (833 counts) in the paper are not comparable with our counts and as a reason of this they are not comparable with the results of the smaller counts in Pfeifer et al. (2013) (see at the end of page 5 of Höller 2017*).**

**I suppose you mean 12 accidents with more than 3 fatalities (see Table 4 of the paper); regarding the influence of "multi-fatalities", please see below.**

**In the case of off-piste fatalities (p=0.055) our data additionally include data from 1977/78 up to 1979/80 and as a result of this we observe different results**

**The Mann-Kendall test, which Techel and Höller employ in their papers, is only sensitive for MONOTONIC trend profiles. But our assumption is that the trend functions are possibly nonlinear and/or not monotonic. In general (and as a consequence of our paper), we think that it is necessary to take nonlinear functions into consideration.**

**Please see lines 278 ff. of the new version of the paper.**

**Finally we agree with Höller (2017) that there is no increase of off-piste fatalities "at the moment". We rather observe a decrease of off-piste fatalities in the most recent years.**

As already pointed out in one of the initial reviews, the influence of multi-fatality accidents on the absolute annual number of fatalities and on the trend function should be discussed. This may be particularly relevant, as the dataset is split into two categories, with low counts for the off-piste category. Although (Höller, 2017) noted no trend in the number of accidents with many fatalities, single events claiming many lives occurred repeatedly (for instance a backcountry accident in 1982 claimed 12 lives).

As said in a previous reply to the reviews, we do not think that single extreme events have an effect on the (nonlinear) estimates because of their robustness (is the case only for a "cluster" of larger Austrian total fatality counts in the mid 80s).

The number of accidents with more than one fatalities is rather decreasing (in case of backcountry significantly linear decreasing; see figure below and AIC/BIC values in the paper)!

[Figure]

[Figure]

Therefore, I suggest to additionally explore the trend for the period 1980/81 until 2015/16 using just the counts of fatal accidents in Austria (in this case, multi-fatality accidents have no influence on annual fatality numbers).

**In case of total fatalities (Fig. 1 of the paper) we are not able to gather the number of avalanches with fatalities beginning from 1967/68 for our purposes (only the number of fatalities).**
**In case of off-piste fatal avalanches we are able to report the number of fatalities beginning from 1980/81 (the first 3 years missing). As you can see the shape of the estimated function is similar (or even more significant because of the decrease of multi-fatal avalanches, see figure above).**

**We did these examinations earlier too, but we decided to take only the number of fatalities which are common in literature. However we are not able to do this in case of other countries (and partly in case of Austria  if we consider total fatal avalanches).**

**We will address this in the discussion!**

**(In a former version of the paper we had a map with numbers of fatal off-piste avalanches, but we skipped this because of lack of space)**

Additionally, the authors could show trend curves and statistics combining the annual backcountry fatalities for the four Alpine countries Austria, France, Italy, Switzerland. From my perspective, these two approaches would considerably strengthen the analysis, allow a more robust interpretation of the results (particularly if trends are confirmed using the different datasets), and would allow a more in-depth-discussion of the advantages and limitations of the statistical model.

[Figure]

**AUT, CHE, FRA, ITA**

Total      Off-piste

**Please see Figure 7 in the new version of the article.**

Furthermore, Höller (2017) provides an extensive overview of publications concerning trends and statistics of avalanche fatalities (Switzerland, USA, Italy, France), some of which might be suitable references when discussing trends.

**Thank you for this advice; we will add the references Pfeifer et. al. (2013) and Höller (2017). However, some other references relating to Switzerland,...,USA are not suitable because of the different data bases, see e.g. (*) above.**

Data and Methods
As you analyze fatality counts only, you could remove line 83 (number of persons involved).

**If we put Figure 1 of this reply (fatal avalanches with fatal.>1) into the paper, we propose not to delete this line.**

Results - Section 3.1 and 3.2
These two sections introduce the figures and tables, but not the results themselves (this was already pointed out in one of the first reviews). Results are presented and discussed in Section 4 (Discussion) only.
I find this a rather unusual approach. I recommend to introduce the figures together with the results they show.

**The results are introduced or presented in the "Results" section and discussed in the "Discussion" section, please see discussion in a former reply (however, we are open for changes if requested by the editor).**

Discussion - Section 4.1
lines 214 to 219: You discuss the increased number of fatalities in the 1980's and mention increased snowfall (in Austria, I assume) but cite two sources which explored Swiss snowfall trends (Abegg, 1998; Laternser and Schneebeli, 2003). Furthermore, you note no peak of off- piste fatalities in the 1980's. As already outlined in one of the initial reviews, these two statements seem contradictory.

**This is literature referring to snow fall trends in the ALPS. In a former version of the article we had a time series plot with snow fall in the St. Anton cluster, see Fig (Solid precipitation during wintertime around St. Anton a. Arlberg within 1900 and 2003 based on precipitation data of the 'Zentralanstalt für Meteorologie und Geodynamik' (ZAMG)):**

[Figure]

**As pointed out in a former reply we have no idea where there is no peak in the off-piste case. (We will address this contradiction in the paper more clearly; an explanation could be: higher frequencies of "multi-fatality" accidents in the case of backcountry skiers).**

lines 264-267: As you point out, AIC/BIC statistics indicate that the curves are non-linear in most cases. Maybe you could formulate more clearly whether this result confirms or contradicts the cited study by Techel et al. (2016) (this did not become clear to me reading these lines).

**As mentioned in a previous reply the data where the results of Techel et. al (2016) are based on (uncontrolled terrain), are different from ours.**
**In Switzerland we notice a decrease in the 1990s and slight increase in the 2000s - see figure 1 (a), Techel et al 2016 and figure 2 (b,d) in case of Switzerland, Austria and Slovenia - which is similar to our case.**
**Unfortunately, there is some variation because of Techel's running mean approach. Here you see our GAM estimate of uncontrolled fatalities in Switzerland (source: data we kindly received from F. Techel, before 1983/84 taken from Figure 1(a), Techel et al 2016):**

[Figure]

Switzerland uncontrolled

Discussion - Section 4.2
I could not find any information whether, and to what extent, the spatial simulation matches the actual observed number of fatalities (the maps show this, but they are hard to interpret - printed numbers vs. background colors). Please show the correlation between observations and simulations, or some other measure of similarity.

**Rsquare adjusted: total 0.972, off-piste 0.954**
**or**
**Deviance explained: total 91.2% off-piste 87.1%**
**see lines 203,204 in the paper.**

Similarly, you could present the statistical correlation between the fatality numbers at municipal level with the proportion of Alpine terrain or overnight tourists. This would facilitate the interpretation for the reader.

**Corr off-piste: 0.66; total: 0.62, see lines 215, 216 in the paper**

**(Alpine terrain: off-piste: 0.27; total: 0.42)**

From my perspective, you should also discuss the benefits and limits of the chosen Markov Random Fields method used for the spatial analysis.

**In a further step a temporal and spatial modeling en bloc (spatio-temporal model) would be nice from a statistical point of view.**

Lines 209 - 215, Figure 1
I personally would interpret Figure 1 as a considerable increase between 1969/70 and 1985/86 (probably significant), with only minor changes afterwards (slight increase until 2005, but 90% confidence interval is large in comparison). This interpretation would correspond quite closely to Höller's results and conclusions, and would also agree quite closely with the trends shown for Switzerland (study you already cite (Techel et al., 2016)). I suggest to show whether fatality counts in the years surrounding 1985 and 2005 differ significantly.

**We would describe it as an increase up to the mid 2000s with a peak in the 1980s (possibly due to increased snowfall in the 80s).**

Figures - 7, 8, 9
The color of the individual polygons in Figures 7 and 8 (simulated number of fatalities) show different information than in Figure 9 (Proportion Alpine terrain). This is not fully intuitive. Therefore, I suggest using different color schemes.

**We changed the colors, see the new version of the article.**

The figure captions (Fig. 7 and 8) miss the information what the background color shows and what the numbers are. This information, currently in the text section 3.2, should be added (or moved) to the caption.

**Thank you for this advice, see modified Figures 9 and 10 in the new paper.**

Please indicate the clusters CL1 and CL2 on the maps in Fig. 7 and Fig. 8 (you refer to them in the text, but non- Austrians will likely not know which location is which).

**We tried this in an earlier version, but we did not find it very nice (such as the borders of the federal states). You should be able to identify the clusters with the accident numbers in the text (I admit: This is a rather tedious task).**

**Reviewer #2:**

Specific comments
Reference 3 (Brugger H, Durrer B, et al. 2001) is outdated. Please use instead "Resuscitation of avalanche victims: Evidence-based guidelines of the international commission for mountain emergency medicine (ICAR MEDCOM): intended for physicians and other advanced life support personnel. Brugger H, Durrer B, Elsensohn F, Paal P, Strapazzon G, Winterberger E, Zafren K, Boyd J; ICAR MEDCOM. Resuscitation. 2013 May;84(5):539-46. doi:10.1016/j.resuscitation.2012.10.020."

**Unfortunately, this article does not refer to the trend of avalanche fatalities over time (such as Brugger et al. 2001).**

**Editor:**

Comments to the Author:
Dear Authors,

Finally, I received two out of three referee reports, and in order not to further delay the publication I made my decisions based on these two reviews. As you can see, referee #2 has no concern about your manuscript, which is fine.
Referee #1, in contrast, raised some additional questions which are worth to be discussed a bit deeper:
- One of the co-authors, Peter Höller, recently published an analysis of Austrian avalanche fatalities in "Cold Regions Science and Technology" entitled "Avalanche accidents and fatalities in Austria since 1946/47 with special regard to tourist avalanches in the period 1981/82 to 2015/16". I am wondering why this study is not referred to in your work, in particular since there are some similarities between the data sets used.

- Furthermore, I particularly would like to see a discussion on the items discussed by referee #2, in particular the contradictions between your data and Höller's published work, as (and here I entirely agree with the referee) discussed in the review.
- More information is given in the extensive comments of reviewer #1.
Please proceed accordingly, and provide a step-by-step answer to the referees' comments when re-submitting your work.

**Please see our statements to Reviewer #1 (I hope you mean "referee #1" instead of "referee #2" in the last paragraph).**

**Last but not least, I would like to thank the reviewers for their helpful comments which helped us to improve our paper considerably!**